# Magnetically steered cell therapy for reduction of intraocular pressure as a treatment strategy for open-angle glaucoma

M Reza Bahranifard[1], Jessica Chan[2], A Thomas Read[3], Guorong Li[4], Lin Cheng[5,6], Babak N Safa[3,7], Seyed Mohammad Siadat[3], Anamik Jhunjhunwala[3], Hans E Grossniklaus[8], Stanislav Y Emelianov[3,9], W Daniel Stamer[4,10], Markus H Kuehn[5,6], C Ross Ethier[1,3]*

[1]Woodruff School of Mechanical Engineering, Georgia Institute of Technology, Atlanta, United States; [2]School of Chemistry and Biochemistry, Georgia Institute of Technology, Atlanta, United States; [3]Wallace H. Coulter Department of Biomedical Engineering, Georgia Institute of Technology/Emory University, Atlanta, United States; [4]Department of Ophthalmology, Duke University, Durham, United States; [5]Department of Ophthalmology and Visual Sciences, The University of Iowa, Iowa City, United States; [6]Veterans Administration Center for the Prevention and Treatment of Visual Loss, Iowa City VA Healthcare System, Iowa City, United States; [7]Department of Medical Engineering, University of South Florida, Tampa, United States; [8]Departments of Ophthalmology and Pathology, Emory University School of Medicine, Atlanta, United States; [9]School of Electrical and Computer Engineering, Georgia Institute of Technology, Atlanta, United States; [10]Department of Biomedical Engineering, Duke University, Durham, United States

*For correspondence:
ross.ethier@bme.gatech.edu

Competing interest: The authors declare that no competing interests exist.

## eLife Assessment

This study has **fundamental** findings that support the potential application of exogenous stem cell therapy as a viable therapeutic option for the management of intraocular pressure (IOP) and to increase outflow facility. The evidence supporting the clinical application of stem cells is **compelling**, using a combination of established in vivo and ex vivo experimental techniques. The work will be of interest to both basic stem cell biologists and clinical glaucoma specialists.

**Abstract** Trabecular meshwork (TM) cell therapy has been proposed as a next-generation treatment for elevated intraocular pressure (IOP) in glaucoma, the most common cause of irreversible blindness. Using a magnetic cell steering technique with excellent efficiency and tissue-specific targeting, we delivered two types of cells into a mouse model of glaucoma: either human adipose-derived mesenchymal stem cells (hAMSCs) or induced pluripotent cell derivatives (iPSC-TM cells). We observed a 4.5 [3.1, 6.0] mmHg or 27% reduction in intraocular pressure (IOP) for 9 months after a single dose of only 1500 magnetically steered hAMSCs, explained by increased outflow through the conventional pathway and associated with a higher TM cellularity. iPSC-TM cells were also effective, but less so, showing only a 1.9 [0.4, 3.3] mmHg or 13% IOP reduction and increased risk of tumorigenicity. In both cases, injected cells remained detectable in the iridocorneal angle 3 weeks post-transplantation. Based on the locations of the delivered cells, the mechanism of IOP lowering

is most likely paracrine signaling. We conclude that magnetically steered hAMSC cell therapy has potential for long-term treatment of ocular hypertension in glaucoma.

## Introduction

Glaucoma, an optic neuropathy, is the leading cause of irreversible blindness, with more than 80 million cases worldwide (*Tham et al., 2014*). Primary open-angle glaucoma (POAG), the most common subtype of the disease, is characterized by a gradual loss of retinal ganglion cells and a corresponding loss of vision. While the exact mechanism underlying retinal ganglion loss is not well understood, elevated intraocular pressure (IOP) is a major risk factor *Coleman and Miglior, 2008*; consequently, all current clinical treatments seek to sustainably lower IOP, using pharmacological, laser, and surgical means. However, the success of such IOP-lowering treatments is reduced by low patient adherence to medical therapies (*Reardon et al., 2011*), by postsurgical complications, and/or by patients becoming refractory to originally successful treatments (*Heijl et al., 2002*). Thus, there remains a major unmet public health need for methods that offer sustained IOP control in glaucoma patients.

The trabecular meshwork (TM; *Figure 1*) is an ocular tissue that drains the majority of aqueous humor (AH) from the human eye, and its function is a major determinant of IOP. There are a number of age- and glaucoma-associated changes in the TM, including an age-associated loss of TM cells which is accelerated in POAG (*Alvarado et al., 1984*). This cell deficiency has been identified as a therapeutic target for IOP control in glaucoma patients, with multiple groups attempting to re-functionalize the TM by injection of stem cells into the eye to restore normal IOP homeostasis (*Manuguerra-Gagné et al., 2013*; *Roubeix et al., 2015*; *Abu-Hassan et al., 2015*; *Zhu et al., 2016*; *Zhu et al., 2017*; *Yun et al., 2018*; *Zhu et al., 2020*; *Zhou et al., 2020*; *Xiong et al., 2021*).

Despite the potential of stem cell treatment for IOP control, there remain several critical barriers to translation. For example, cell delivery to the TM has typically relied on passive transport of cells by AH outflow, leading to extremely low delivery efficiencies (*Wang et al., 2022*). A more efficient delivery method is desirable, which is expected to both increase the therapeutic benefit of the treatment and reduce immunogenicity; e.g., *Zhou et al., 2020* reported an increase in the inflammatory markers CD45 and GR1 and T-cell markers CD4 and CD3 in the iris and the cornea after mesenchymal stem cell injection, likely due to off-target cell delivery. We have recently introduced a magnetically steered cell delivery technique which significantly outperforms previously used magnetic and nonmagnetic delivery techniques (*Bahrani Fard et al., 2023*). Here, we characterize the efficacy of our stem cell delivery using this technique to lower IOP.

A second barrier to translation is the lack of knowledge about which cell type should be delivered to restore TM function. Three types of cells have previously been used in this context: native TM stem cells (TMSCs), mesenchymal stem cells, and induced pluripotent stem cell (iPSC) derivatives. TMSC therapy lowers IOP and increases TM cellularity (*Yun et al., 2018*; *Xiong et al., 2021*) and is theoretically attractive. However, the scarcity of TMSCs, constituting only 2–5% of the entire TM cell population (*Braunger et al., 2014*), and the invasiveness of the required cell collection procedure significantly reduce the translational potential of this cell source. Alternatively, mesenchymal stem cells have been used in several studies, showing a transient IOP reduction, as well as neuroprotection (*Manuguerra-Gagné et al., 2013*; *Roubeix et al., 2015*). For example, Manuguerra-Gagné et al. injected bone-marrow-derived mesenchymal stem cells in a rat model of IOP elevation, observing a reduction in IOP for 3 weeks (*Manuguerra-Gagné et al., 2013*). These results, together with the ease of sourcing autologous cells and the established safety of mesenchymal stem cell therapy in clinical trials (*Lalu et al., 2012*; *Rodríguez-Fuentes et al., 2021*), make these stem cells a strong candidate for clinical POAG cell therapy. Finally, iPSCs can be differentiated into iPSC-TM cells, with the differentiated cells displaying phenotypic similarity to adult TM cells (*Ding et al., 2014*). Intracameral injection of iPSC-TMs into a perfused porcine anterior segment POAG model restored the IOP homeostatic response (*Abu-Hassan et al., 2015*). Additionally, Zhu and colleagues delivered iPSC-TMs into the anterior chambers of ocular hypertensive mice and reported increased TM cellularity due to proliferation of endogenous cells and a corresponding decrease in the IOP for up to 12 weeks after cell delivery (*Zhu et al., 2016*; *Zhu et al., 2017*). Although both MSC and IPSC-TM showed promise, there is no information about their efficacy and safety in comparison to each other. Therefore, here, we compare the benefits of mesenchymal stem cells vs. iPSC-TM cells.

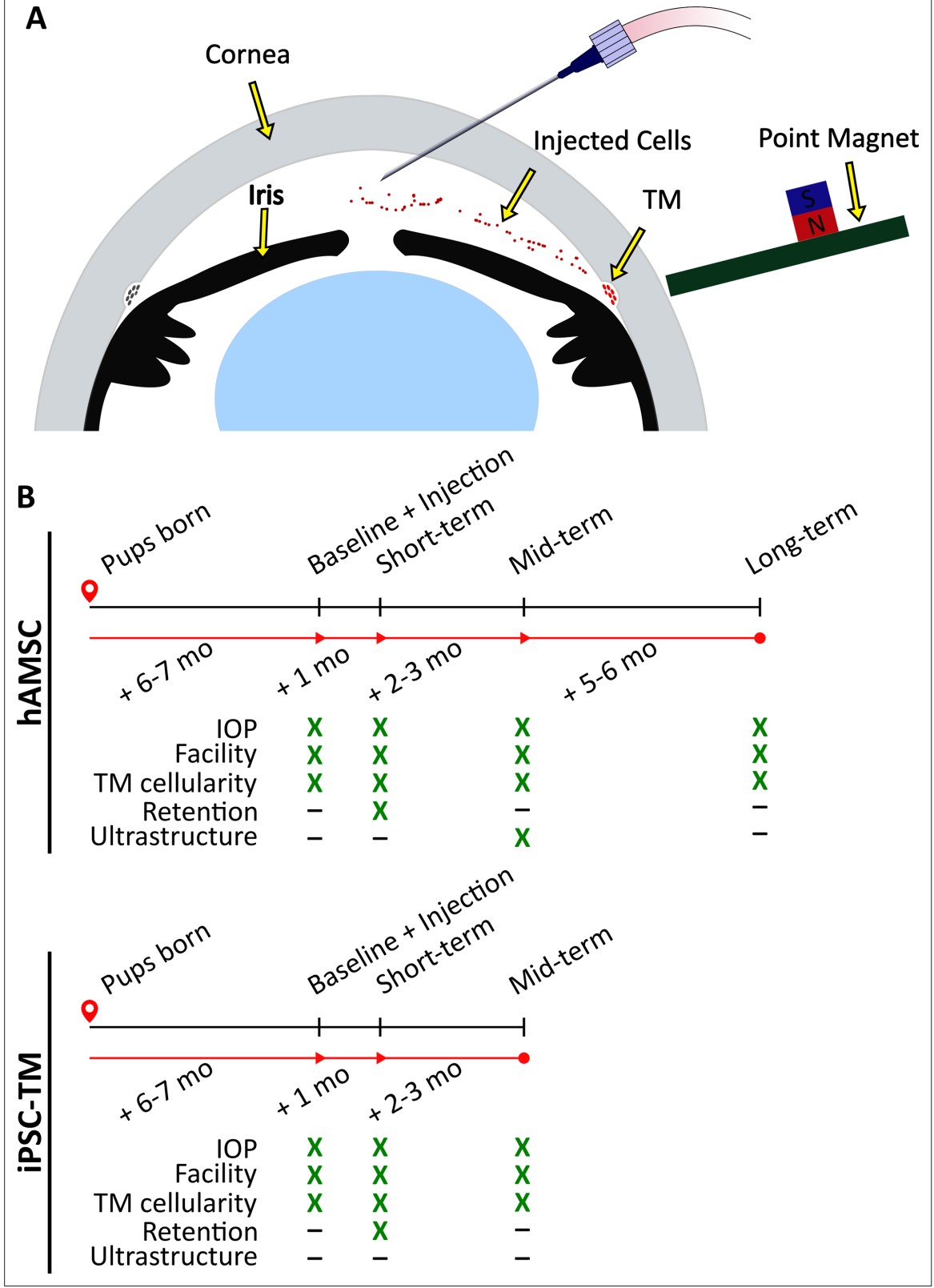

**Figure 1.** Experimental schematic and design. (**A**) Schematic of magnetically steered cell delivery to the trabecular meshwork (TM). As cells are injected into the anterior chamber at a low flow rate, the experimenter places the 'point magnet' (*Bahrani Fard et al., 2023*) on the limbus and carefully drags the cells toward the iridocorneal angle, targeting the TM. Features of the figure are not drawn to scale. (**B**) Timeline of the experiments. An ultrastructural analysis, specifically the quantification of inner wall basement membrane fenestrations, was not undertaken for eyes receiving induced

*Figure 1 continued on next page*

**Figure 1 continued**

pluripotent cell derivatives (iPSC-TMs) due to their inferior performance. Additionally, cell retention in the anterior chamber was only investigated at the short term. Note that baseline measurements were taken for wild-type (WT) and transgenic animals that did not necessarily receive an injection afterward. Refer to Materials and methods for a description of various experimental groups and further details.

An additional barrier to translation is the choice of an appropriate animal model for preclinical testing, since no animal model replicates all the pathological phenotypes of POAG. For example, although nonhuman primate models show high anatomical and functional resemblance to humans and are the gold standard for certain preclinical studies (*Friedman et al., 2017*), induction of ocular hypertension requires laser photocoagulation of the TM, which is very unlike TM changes seen in POAG (*Manuguerra-Gagné et al., 2013*; *Acott et al., 1989*). Microbead (*Sappington et al., 2010*) and hypertonic saline (*Morrison et al., 1997*) models of ocular hypertension are similarly distinct from human POAG pathology. Commercially available DBA/2 mice show TM cell loss and IOP elevation but are associated with undesirable systemic and ocular complications (*Turner et al., 2017*). Thus, in this work, we chose to use transgenic MYOC$^{Y437H}$ mice. These mice carry a glaucoma-causing point mutation in the MYOC gene and have been reported to show an accelerated loss of TM cellularity and a gradual ocular hypertension development (*Zode et al., 2011*).

In summary, we here evaluate the effectiveness of TM cell therapy, using a magnetic cell steering method and two clinically relevant cell choices, namely human adipose-derived mesenchymal stem cells (hAMSCs) and iPSC-TM cells, in MYOC$^{Y437H}$ mice. We judged effectiveness by the extent and longevity of IOP reduction, improvement in outflow facility, and increase in TM cellularity, among other outcome measures, and performed experiments using animal cohorts at different time points (*Figure 1B*, detailed in Materials and methods section). Despite the absence of ocular hypertension in our MYOC$^{Y437H}$ mice, our data demonstrate sustained IOP lowering and a significant benefit of magnetic cell steering in the eye, particularly for hAMSCs, strongly indicating further translational potential.

## Results

### Cell transplantation lowered IOP and improved AH dynamics

We delivered and magnetically steered either hAMSCs or iPSC-TMs to the TM of eyes of MYOC$^{Y437H}$ mice (*Figure 1*; mice described above), measuring IOPs and outflow facilities at short-, mid-, and long-term time points, corresponding to ~1 month, 3–4 months, and 9 months after cell delivery (*Figure 1*). Our high-level goals were to: (i) elucidate the impact of hAMSC or iPSC-TM delivery on IOP; and (ii) quantify the portion of IOP change due to changes in outflow facility, a key functional metric of the TM. Outflow facility is the numerical inverse of the hydraulic resistance to AH drainage through the conventional outflow pathway.

We expected transgenic MYOC$^{Y437H}$ mice to show elevated IOP by 6–7 months of age, when baseline IOP measurements were taken (*Zhu et al., 2016*; *Zhu et al., 2017*; *Zode et al., 2011*). Surprisingly, we saw no meaningful IOP difference between Tg-MYOCY437H mice (Tg group) and wild-type (WT) littermates (*Figure 2*; *Table 1* and *Table 2*). Despite this lack of IOP elevation in the transgenic model, magnetically steered delivery of hAMSCs led to a marked IOP decrease in Tg animals as compared to sham injection of saline at short-, mid-, and long-term time points. The IOP reduction was sustained in hAMSC-treated eyes over all three time points, with no statistically significant difference between any combination of these time points. iPSC-TM treatment also led to a reduction in IOP compared to sham (phosphate-buffered saline [PBS]) injection controls at both the short- and mid-term time points, although this difference did not reach statistical significance at the latter time. The IOP reduction due to iPSC-TM cells was approximately half that due to hAMSC treatment at both short- and medium-term time points (short term: $-4.3 \left[-5.6, -2.9\right]$ mmHg for hAMSC vs. $-2.3 \left[-3.6, -1.0\right]$ mmHg for iPSC-TM, $p = 0.021$; mid-term: $-4.5 \left[-5.8, -3.1\right]$ mmHg for hAMSC vs. $-1.9 \left[-3.3, -0.4\right]$ mmHg for iPSC-TM, $p = 0.005$; all data reported as means and 95% confidence intervals).

We observed increases in outflow facility for eyes receiving stem cells which were consistent with observed changes in IOP (*Figure 3A and B*; *Table 1* and *Table 2*). Specifically, no significant difference in facility was found between the naïve WT and transgenic groups, while hAMSC treatment led to a marked increase in outflow facility vs. injection (sham) controls at short-, mid-, and long-term time

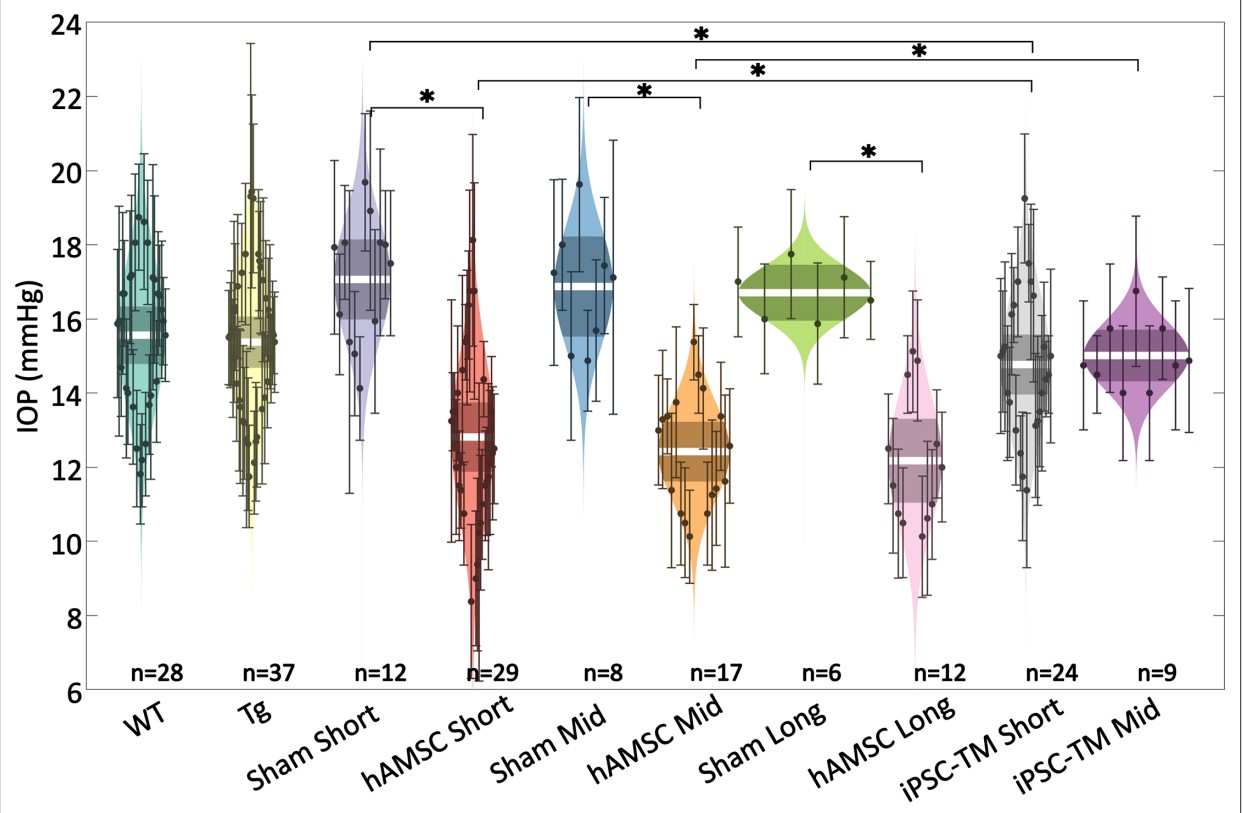

**Figure 2.** Intraocular pressure (IOP) was reduced by targeted cell delivery to the trabecular meshwork (TM). In each experimental cohort, the central white strip indicates the mean, while the darker region represents the 95% confidence interval on the mean. The colored region shows the distribution. Dots represent individual eyes, with error bars demarcating the 95% confidence intervals. For further information on experimental groups and statistical analysis, refer to text. $*p < 0.05$ with Bonferroni correction (see *Table 2*). See below for a more complete statistical analysis. WT: wild-type hybrid mice (naïve control), Tg: Tg- MYOCY437H mice, sham: Tg mice receiving saline injection, human adipose-derived mesenchymal stem cell (hAMSC): Tg mice receiving magnetically steered hAMSCs, induced pluripotent cell derivative (iPSC-TM): Tg mice receiving magnetically steered iPSC-TMs. 'Short', 'mid', and 'long' refer to time points. n is the number of eyes measured in each group.

points. Further, the percentage increases in facility due to hAMSC delivery vs. sham injection controls were similar at all time points. Groups receiving iPSC-TMs also showed an increase in facility, but these differences did not reach statistical significance. Specifically, hAMSC delivery led to a significantly higher percentage increase in facility compared to iPSC-TM delivery: short-term: 170 [70, 310] %

**Table 1.** Outcome measures, shown as means and [95% confidence intervals].

| Group | IOP (mmHg) | Facility (nl/min/mmHg) | Cellular density (nuclei/μm) |
|---|---|---|---|
| WT | 15.6 [14.8,16.3] | 4.4 [3.7,5.2] | 0.24 [0.15,0.33] |
| Tg | 15.4 [14.7,16.1] | 3.8 [3.3,4.4] | 0.28 [0.23,0.32] |
| Sham Short | 17.1 [16.0,18.1] | 3.2 [2.3,4.3] | 0.27 [0.20,0.34] |
| MSC Short | 12.8 [11.9,13.8] | 8.4 [6.3,11.2] | 0.58 [0.42,0.73] |
| Sham Mid | 16.9 [15.5,18.2] | 3.0 [2.1,4.3] | 0.24 [0.18,0.31] |
| MSC Mid | 12.4 [11.6,13.2] | 8.4 [7.1,9.9] | 0.40 [0.34,0.47] |
| Sham Long | 16.7 [16.0,17.5] | 3.4 [2.8,4.2] | 0.23 [0.17,0.28] |
| MSC Long | 12.2 [11.0,13.3] | 8.0 [5.9,10.9] | 0.37 [0.31,0.43] |
| iPSC-TM Short | 14.8 [14.0,15.6] | 4.3 [3.0,6.2] | 0.36 [0.27,0.45] |
| iPSC-TM Mid | 15.0 [14.3,15.7] | 4.3 [3.6,5.1] | 0.34 [0.21,0.47] |

**Table 2.** Result of multiple comparisons for various groups and variables, with statistically significant comparisons highlighted in orange*.

| | Compared groups | | p-Values | | |
| --- | --- | --- | --- | --- | --- |
| | | vs. | IOP | Facility | Cellular density |
| 1 | WT | Tg | 0.3579 | 0.1353 | 0.6264 |
| 2 | Tg | Sham Short | 0.0070 | 0.0901 | 0.9214 |
| 3 | Tg | Sham Mid | 0.0311 | 0.0614 | 0.2879 |
| 4 | Tg | Sham Long | 0.0652 | 0.1865 | 0.2442 |
| 5 | Tg | hAMSC Short | <0.0001 | <0.0001 | <0.0001 |
| 6 | Tg | hAMSC Mid | <0.0001 | <0.0001 | 0.0003 |
| 7 | Tg | hAMSC Mid | <0.0001 | <0.0001 | 0.008 |
| 8 | Tg | iPSC-TM Short | 0.1287 | 0.2044 | 0.0239 |
| 9 | Tg | iPSC-TM Mid | 0.3073 | 0.1296 | 0.0303 |
| 10 | Sham Short | hAMSC Short | <0.0001 | <0.0001 | <0.0001 |
| 11 | Sham Short | iPSC-TM Short | 0.0006 | 0.0692 | 0.0535 |
| 12 | hAMSC Short | hAMSC Mid | 0.2786 | 0.4838 | <0.0001 |
| 13 | hAMSC Short | hAMSC Long | 0.2111 | 0.4045 | <0.0001 |
| 14 | hAMSC Short | iPSC-TM Short | 0.0013 | 0.0019 | <0.0001 |
| 15 | Sham Mid | hAMSC Mid | <0.0001 | <0.0001 | <0.0001 |
| 16 | Sham Mid | iPSC-TM Mid | 0.0047 | 0.0210 | 0.0072 |
| 17 | hAMSC Mid | hAMSC Long | 0.3497 | 0.3719 | 0.5595 |
| 18 | hAMSC Mid | iPSC-TM Mid | <0.0001 | <0.0001 | 0.2751 |
| 19 | Sham Long | hAMSC Long | <0.0001 | <0.0001 | 0.0020 |
| 20 | iPSC-TM Short | iPSC-TM Mid | 0.3597 | 0.4877 | 0.8516 |

*Post hoc comparisons were performed after ANOVA (for IOP and outflow facility) or linear mixed-effect model (for TM cellularity). Bonferroni correction was used to adjust the critical p-value from 0.05 to 0.0025 (based on the 20 reported comparisons).

for hAMSC vs. $40 \left[-10, 110\right]$ % for iPSC-TM, $p = 0.011$ ; mid-term: $180 \left[110, 280\right]$ % for hAMSC vs. $40 \left[0, 110\right]$ % for iPSC-TM, $p = 0.003$; data reported as percent increase in treatment group compared to relevant (sham) injection control.

We then asked whether the measured decreases in IOP were quantitatively consistent with the experimentally measured increases in outflow facility. To answer this question, we used the modified Goldmann equation (**Brubaker, 2004**), which relates IOP to facility and other variables, computing an 'expected' IOP from the facility measurements for each cohort of mice. Comparison of this expected IOP with the actual (measured) IOP showed a close correlation (**Figure 3Ci**), determined by linear regression (slope of fitted line was not statistically different from one, $p = 0.22$, $R^2 = 0.99$). Still, the outflow facility measurements overestimated the actual IOP by a small amount ($1.2 \left[1.1, 1.3\right]$ mmHg, $p < 10^{-6}$, null hypothesis: average difference between experimental and expected IOPs equals zero, **Figure 3Cii**). Despite this 'shift' between the experimentally measured and expected IOP, the horizontal error bars in **Figure 3Ci**, derived by a propagation of error analysis, include the unity line for all groups, suggesting that the small discrepancy between the two experimental and expected values falls within the measurement errors (see Discussion).

## Cell delivery increased TM cellularity

The observed reductions in IOP and increases in outflow facility after delivery of both cell types suggested functional changes in the conventional outflow pathway. We therefore asked whether

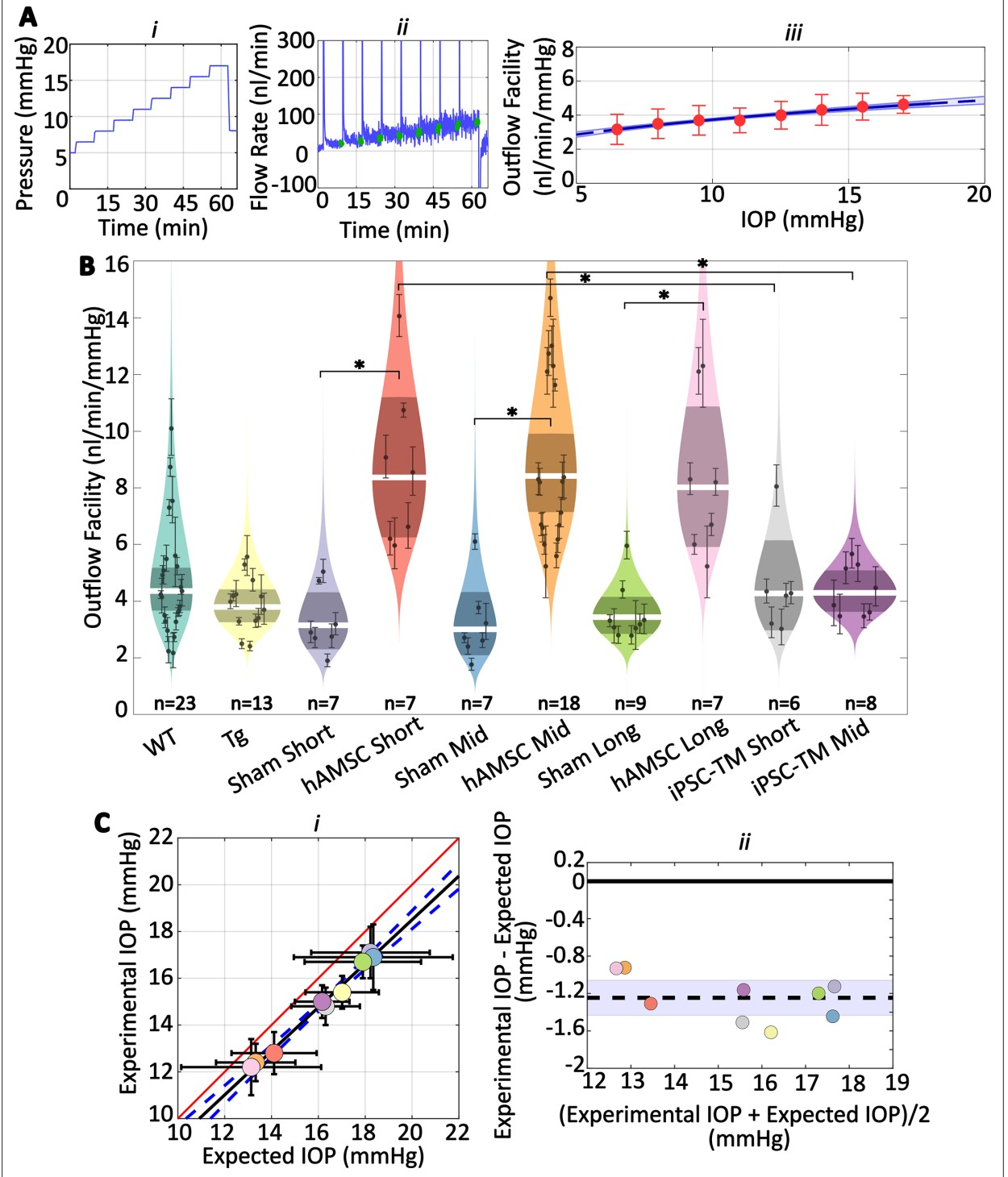

**Figure 3.** Human adipose-derived mesenchymal stem cells (hAMSCs) increase outflow facility in mice. (**Ai–i**i) Intraocular pressure (IOP) and flow rate vs. time for multiple pressure steps during the perfusion experiment in a representative eye. After each pressure step, the perfusion system automatically waits for a steady-state inflow rate to be achieved, based on the criterion that the rate of change in the inflow rate falls below 3 nl/min/min. The steady intervals for each step are shown in green. Data has been trimmed to not include preparatory and pre-loading phases. (**Aiii**) Calculated outflow facility (red dots) vs. IOP. The solid curve shows the fitted model with the shaded region being the 95% confidence interval on the regression line. Error bars are 95% confidence intervals on individual steps. (**B**) Outflow facilities across different experimental cohorts. Refer to *Figure 2* for interpretation details. Note that outflow facilities in mice follow a log-normal distribution. (**C**) Cross-validation of experimentally measured and expected IOP, calculated from measured facility values. (**i**) Regression plot of experimental vs. expected IOPs. Solid black fitted line ($y = 0.94x - 0.21$, $R^2 = 0.99$) is shown along

*Figure 3 continued on next page*

*Figure 3 continued*

with its confidence bounds in dashed blue. Error bars show the 95% confidence interval on both experimental (vertical) and expected (horizontal) IOPs. The unity line is shown as a solid red line. (ii) Bland-Altman plot of IOP residuals (experimental minus expected IOPs) vs. average of experimental and expected IOPs. Individual experimental groups are indicated by colors matching those in panel B. Dashed line shows the mean and is shown along with its 95% confidence interval (shaded). Solid line shows zero difference between the two parameters, i.e., the null hypothesis. For further information on experimental groups and statistical analysis, refer to text. $p < 0.05$ with Bonferroni correction (see **Table 2**).

these changes were associated with alteration of the cellular density in the TM by evaluating cell counts in histological sections of the iridocorneal angle from all of our experimental groups (**Figure 4**; **Table 1** and **Table 2**). We observed more nuclei in eyes receiving cell transplantation, with a striking 2.2-fold increase in TM cellularity (normalized to the anterior-posterior length of the outflow tissues) after hAMSC treatment at the short-term time point vs. the corresponding (sham) injection control (**Figure 4C**). Interestingly, this spike in TM cell density was followed by a decline over time, reaching a 1.6-fold increase at the mid-term time point and apparently plateauing at 1.6-fold at the long-term time point. Despite this modest decline, hAMSC-treated eyes showed significantly higher cellular density vs. their injection controls at both mid-term and long-term time points.

A potential concern with superparamagnetic iron oxide nanoparticle (SPION) loading of stem cells to enable magnetic steering is that SPIONs could cause stem cell cytotoxicity or could leak from stem cells after delivery to the TM, and thus cause cytotoxicity in TM cells. We therefore searched for SPIONs in eyes after stem cell delivery (Appendix 1). Unfortunately, SPIONs were somewhat difficult to visualize, but we did not see notable SPION accumulation in the TM, a finding that is consistent with the significant increase in TM cellularity after MSC delivery.

Delivery of iPSC-TMs also led to an increase in TM cellular density vs. (sham) injection controls at both short-term and mid-term time points, although these differences were more modest than seen in hAMSC-injected eyes and did not reach statistical significance. Interestingly, the TM cellular densities in iPSC-TM-treated eyes at both time points were comparable to those at the mid-term and long-term time points in hAMSC-treated eyes but were significantly different than hAMSC-treated eyes at the short-term time point.

Cross-plotting normalized TM cellularity vs. IOP for pooled data from all the experimental groups (**Figure 4D**) showed a strong negative correlation between these two parameters, indicating an association between greater TM cellularity and lower IOP.

## hAMSC transplantation significantly decreased basement membrane material

An increased deposition of extracellular matrix (ECM), and, in particular, basement membrane material (BMM), in the TM immediately adjacent to the inner wall (IW) of Schlemm's canal has been associated with ocular hypertension (**Li et al., 2021**; **Overby et al., 2014**). Since this region within the TM accounts for the majority of AH outflow resistance (**Ethier et al., 1986**), we asked whether the amount of BMM was altered by stem cell treatment. To address this question, we compared the mid-term hAMSC transplanted group vs. its corresponding injection control (**Figure 5**), selecting the mid-term time point for this analysis since this was the longest time point previously studied (**Zhu et al., 2017**), and we were interested in persistent ECM changes within the TM.

Reduced amounts of BMM adjacent to the IW of Schlemm's canal were evident in transmission electron micrographs from eyes receiving hAMSCs compared to sham-injected controls at the mid-term time point (**Figure 5A**). Quantification showed that stem cell treatment significantly decreased the amount of BMM, as determined by the ratio of BMM length adjacent to Schlemm's canal IW to total IW length; specifically, this ratio was 0.52 [0.34, 0.70] in sham-treated eyes vs. 0.34 [0.22, 0.47] in hAMSC-treated eyes ($p < 0.0001$). The BMM length ratio was measured by two independent annotators (MRB, CRE), and differences between the annotators were not significant (annotator considered as a random effect, $p \approx 1$, likelihood ratio test).

## Exogenous cells were retained for multiple weeks in the TM

Manuguerra-Gagné et al. previously reported a surprisingly low retention duration of hAMSCs in the TM of rat eyes, with virtually no fluorescently labeled exogenous cells being present in histological sections 4 days after injection (**Manuguerra-Gagné et al., 2013**). We therefore pre-labeled injected

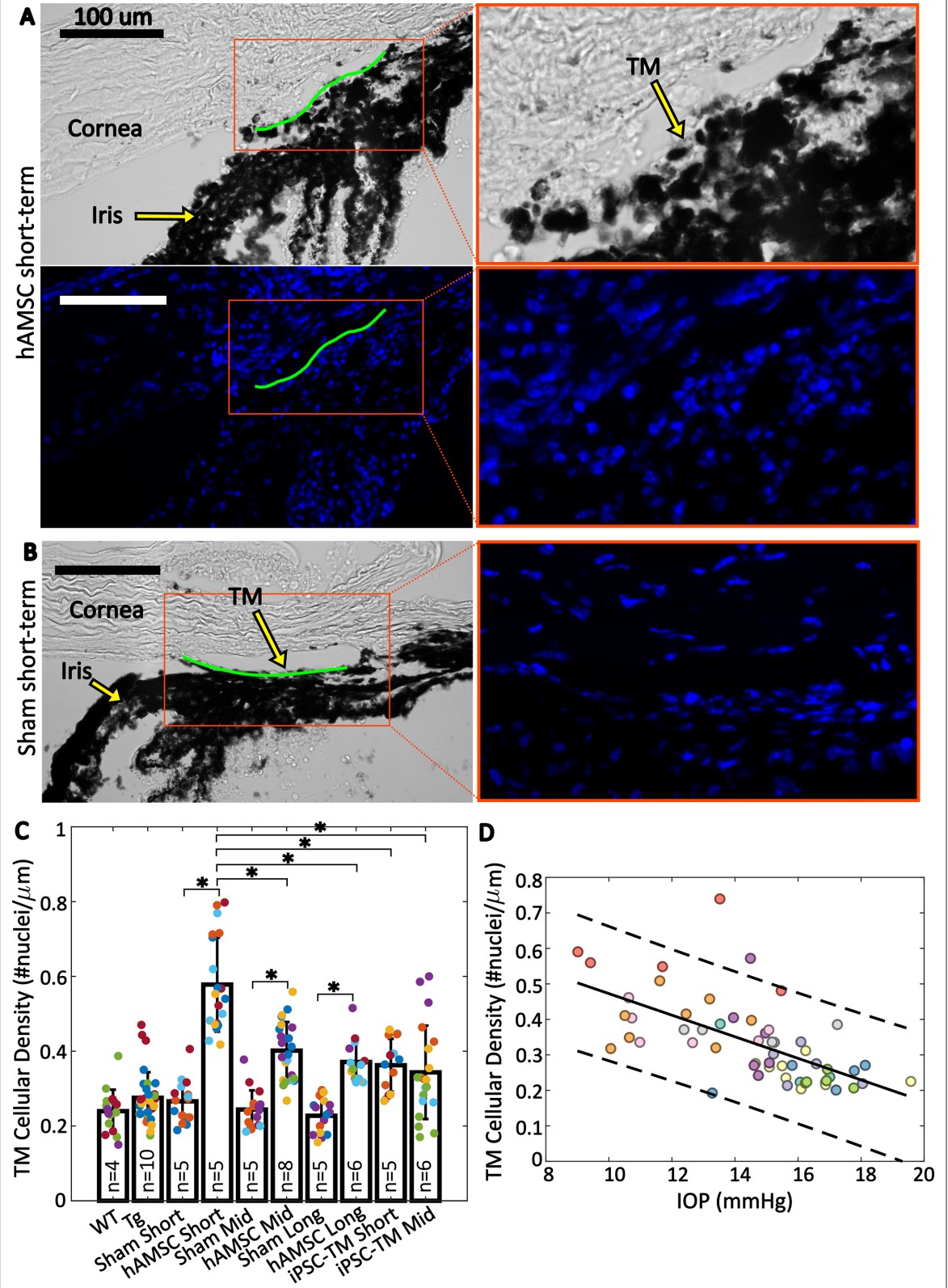

**Figure 4.** Trabecular meshwork (TM) cellular density is improved by stem cell delivery. (**A and B**) Bright-field and fluorescent micrographs of the irideocorneal angle (sagittal view) taken from a representative eye from the hAMSC short-term (**A**) and sham short-term (**B**) experimental groups (see *Figure 4—figure supplement 1* for more details). Green line shows the contour of the TM along the inner wall of Schlemm's canal used for normalizing nuclei count. DAPI-stained nuclei in the fluorescent image are shown in blue. Adjacent panels show a magnified view of the angle. (**C**) Comparison

*Figure 4 continued on next page*

*Figure 4 continued*

of TM cellular density (number of nuclei per length of inner wall of Schlemm's canal) for various experimental cohorts. Bars show mean and standard deviation. Multiple sections analyzed from each eye are coded with the same color. n=number of eyes. Linear mixed-effect model, $*p < 0.05$ with Bonferroni correction (see *Table 2*). (**D**) Cross-comparison of TM cellularity vs. IOP for the eyes shown in panel (**C**). The negative correlation between the variables was statistically confirmed (Pearson's correlation coefficient = –0.63 and $p < 10^{-7}$). Each color represents one eye, with different colors matching the experimental groups as shown in *Figure 2*. Trend line (solid) is shown along with the 95% prediction interval (dashed).

The online version of this article includes the following figure supplement(s) for figure 4:

**Figure supplement 1.** Complementary micrographs to *Figure 4* used for trabecular meshwork (TM) cellularity quantifications.

cells with PKH26 fluorescent dye, which allowed us to track cells for up to 3 weeks after injection. En face images showed a relatively uniform distribution of cells over the entire circumference of the eye (*Figure 6A*), similar to previous results with magnetically steered cells (*Bahrani Fard et al., 2023*). Sagittal sections (*Figure 6B*) showed an accumulation of exogenous cells deep within the iridocorneal angle. Interestingly, strong fluorescent signals were observed within the TM in iPSC-TM-injected eyes, indicating cell integration with the target tissue; in contrast, most hAMSCs accumulated close to the TM (within 50 μm), but did not enter the TM. Note that fluorescent signals observed in the posterior part of the eye and outside the eye near the limbus were caused by autofluorescence (*Figure 6— figure supplement 1*).

## iPSC-TM transplantation led to a significant incidence of tumor formation

Unfortunately, there was a very high rate of tumorigenicity in eyes receiving iPSC-TMs, with more than 60% of eyes showing large intraocular masses within a month of cell injection, typically on the iris (*Figure 6B*). In most cases, these tumors left the eyes unusable for IOP or outflow facility measurements. Examination of select iPSC-TM-transplanted sections by a board-certified pathologist (HEG) confirmed the presence of tumors (*Figure 7*), based on observation of rosettes and neuroectodermal phenotype, characteristics also found in various tumor types, including retinoblastoma (*Wippold and Perry, 2006*). Additionally, a high nuclear-cytoplasmic ratio, a hallmark of tumor malignancy (*Moore et al., 2019*), and rarefaction due to tissue necrosis were noted. No signs of tumor growth were observed in the eyes injected with hAMSCs at the long-term time point (*Figure 7*).

## Discussion

The overarching goal of this study was to evaluate the effectiveness of a magnetic TM cell delivery technique we previously developed (*Bahrani Fard et al., 2023*). Specifically, by delivering stem cells into the eyes of a mutant myocilin mouse model of POAG and observing the effects on IOP and AH dynamics for an extended period of time, we wished to evaluate the potential of this treatment for eventual clinical translation (*Coulon et al., 2022*). We hypothesized that our targeted magnetic delivery approach would prove efficacious. A secondary goal was to compare the efficacy of two clinically relevant stem cell types: hAMSCs and iPSCs that had been differentiated toward a TM cell phenotype (iPSC-TMs).

### hAMSC delivery led to long-term IOP reduction

Our major finding was that magnetically steered delivery of hAMSCs led to a significant and sustained lowering of IOP, which could be almost entirely explained by improved function of the conventional outflow pathway. Specifically, we saw an ~27% (4.5 mmHg) IOP reduction in eyes receiving hAMSCs vs. saline (sham) injection control eyes, which was sustained for 9 months after cell delivery. This lowering of IOP was closely related to a stable ~2.8-fold increase in outflow facility in the hAMSC treatment group vs. saline injection controls.

Eyes subjected to saline injection exhibited marginally higher IOPs and lower outflow facilities on average, in comparison to the transgenic animals at baseline. However, due to the lack of statistical significance in these differences and the inherent age difference between the saline-injected animals and the non-injected controls at baseline, no conclusive inference can be drawn regarding the effect of saline injection. To investigate this phenomenon further, we also analyzed IOPs in uninjected contralateral eyes at the mid-term time point (Appendix 2). The uninjected contralateral transgenic

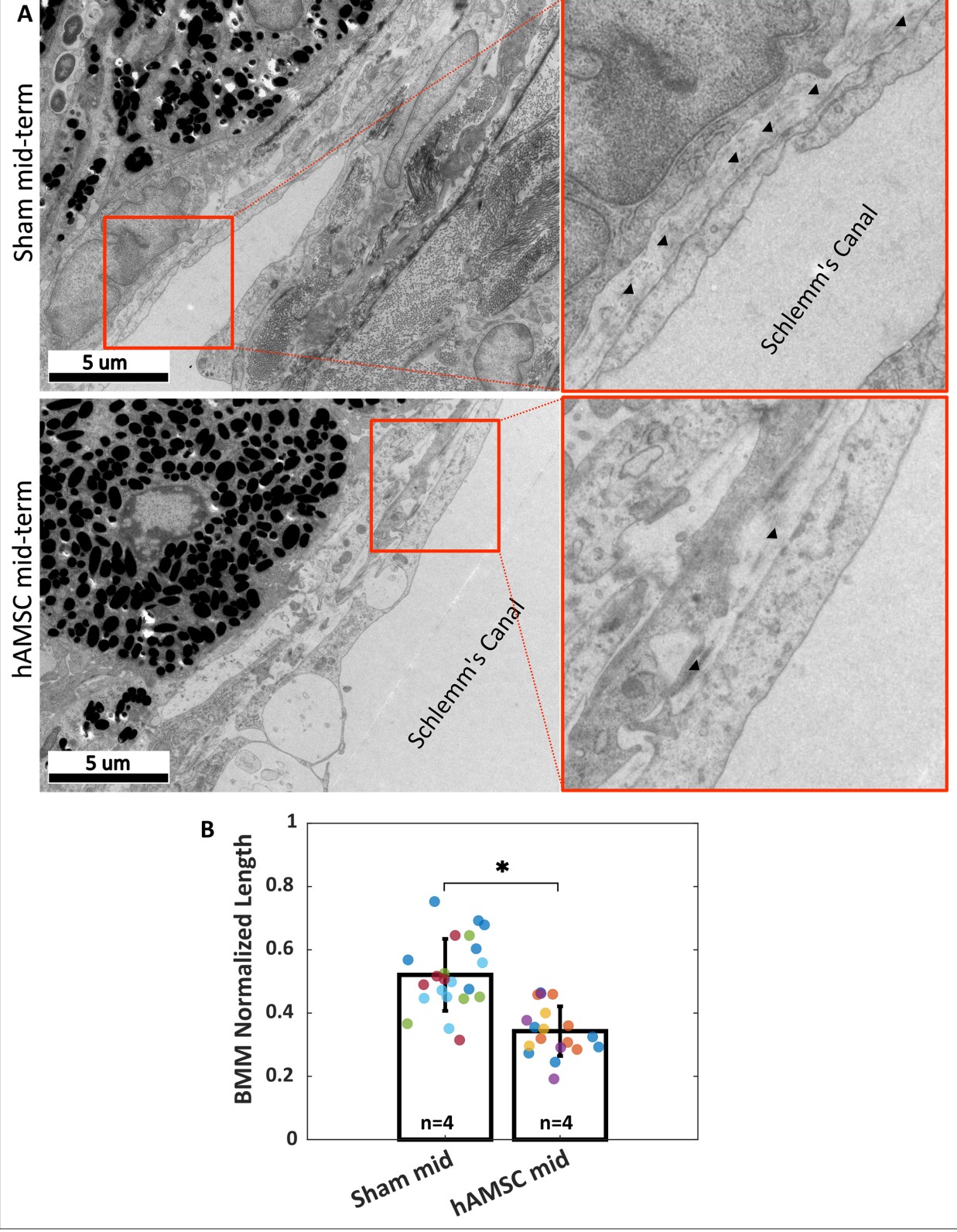

**Figure 5.** Ultrastructural analysis of extracellular matrix (ECM) underlying the inner wall (IW) of Schlemm's canal (basement membrane material [BMM]). (**A**) Greater amounts of BMM are evident immediately adjacent to the IW of Schlemm's canal (arrowheads) in a saline-injected eye (top row) vs. in a human adipose-derived mesenchymal stem cell (hAMSC)-treated eye (bottom row) at the mid-term time point. The images at right are a zoomed view of the orange boxed areas in the left panels. (**B**) The normalized length of BMM directly in contact with the IW (length of BMM material divided by

*Figure 5 continued on next page*

*Figure 5 continued*

length of IW of Schlemm's canal) for the experimental groups represented in panel A. Dots represent the average value between annotators for each measured section. Multiple sections analyzed from each eye are coded with the same color. n=number of eyes. Bars and error bars represent means and standard deviations. Linear mixed-effect model. $p < 0.05$.

The online version of this article includes the following figure supplement(s) for figure 5:

**Figure supplement 1.** Quantification procedure for the amount of basement membrane materials (BMM) underlying to the inner wall of Schlemm's canal.

eyes (10 months of age) showed an IOP of 16.5 [15.9, 17.1] mmHg, which was intermediate between the IOP levels of the 6- to 7-month-old Tg group (15.4 [14.7, 16.1] mmHg) and the sham-injected group (16.9 [15.5, 18.2] mmHg). However, none of these differences reached statistical significance. Of note, contralateral hypertension has been previously reported after subconjunctival and periocular injection of dexamethasone-loaded nanoparticles (*Li et al., 2019*), and we similarly cannot definitively rule out potential contralateral effects induced by our stem cell injections. Thus, we cannot draw any definite conclusions from these additional IOP comparisons at this time.

Our measured IOPs were close to 'expected IOPs' calculated from facility measurements, strongly suggesting that the majority of the IOP-lowering effect after hAMSC delivery was due to an improvement in the function of the conventional outflow pathway.

## There was a slight offset between measured and expected IOPs

Despite the very close correlation between measured and expected IOPs noted above, there was a small but consistent offset between these two quantities, which may be due to several factors. First, cell delivery could theoretically cause a decrease in the rate of AH formation or an increase in the rate of uveoscleral outflow, which would lower experimentally measured IOP. However, according to *Equation 2*, a change in the pressure-independent flow rate ($Q$) would disproportionately affect the IOP in groups with lower facility. For example, if we conservatively assume that the 1.2 mmHg average residual (experimental IOP – expected IOP) was caused by a difference in inflow rate for transgenic animals vs. WT animals, which we used as the reference for calculating inflow rate (see Materials and methods), the 95% confidence interval on the mean of the residuals would have been ~5 times larger than what we actually calculated. The second possible explanation is that the mismatch was caused by an error in rebound tonometry, e.g., due to tonometer miscalibration or an anesthesia-induced drop in IOP (*Qiu et al., 2014*). However, if we assume that all groups, including WT animals, had an experimentally measured IOP that was artifactually lower than true IOP, the pressure-independent flow rate ($Q$) calculated for WT animals would incorporate this effect. Thus, when this $Q$ is used to calculate expected IOPs for groups other than WT animals, there should not be an offset between the experimental and expected IOPs, at least for groups with facilities similar to WT animals. We therefore suggest that the most plausible explanation is an inherent difference between the transgenic and WT animals, such as in the biomechanical properties of the cornea (leading to an error in the IOP read by the tonometer), in the episcleral venous pressure, or in the amount of IOP reduction due to anesthesia.

Despite some uncertainty about the minor offset between the expected and measured IOPs, the data strongly suggests that the IOP lowering caused by stem cell therapy is largely due to a restoration of function to the conventional outflow pathway.

## hAMSC treatment led to increased TM cellularity and reduced BMM

One of the hallmarks of POAG is loss of TM cells (*Alvarado et al., 1984*), which was an early motivation for TM cell therapy as a potential treatment for this disease. We found that hAMSC delivery led to a striking 2.2-fold increase in TM cellularity 3–4 weeks after treatment vs. saline-injected controls, which showed cellularities similar to eyes from WT mice. This increased cellularity declined somewhat by 3–4 months after cell injection but then stabilized for up to 9 months after injection. Additionally, the increase in cellularity was strongly correlated with a decrease in IOP for pooled data from all the experimental groups. This correlation more directly highlights the potential of TM cell therapy in treating ocular hypertension, where TM cellularity is reduced and IOP is elevated (*Alvarado et al., 1984*). Interestingly, *Alvarado et al., 1984* showed that humans at birth have ~2.3-fold higher TM cellularity compared to a 40-year-old individual, and that this cellularity reduces sharply within the first 5 years of life. This trend in human eye cellularity resembles, both qualitatively and quantitatively, our

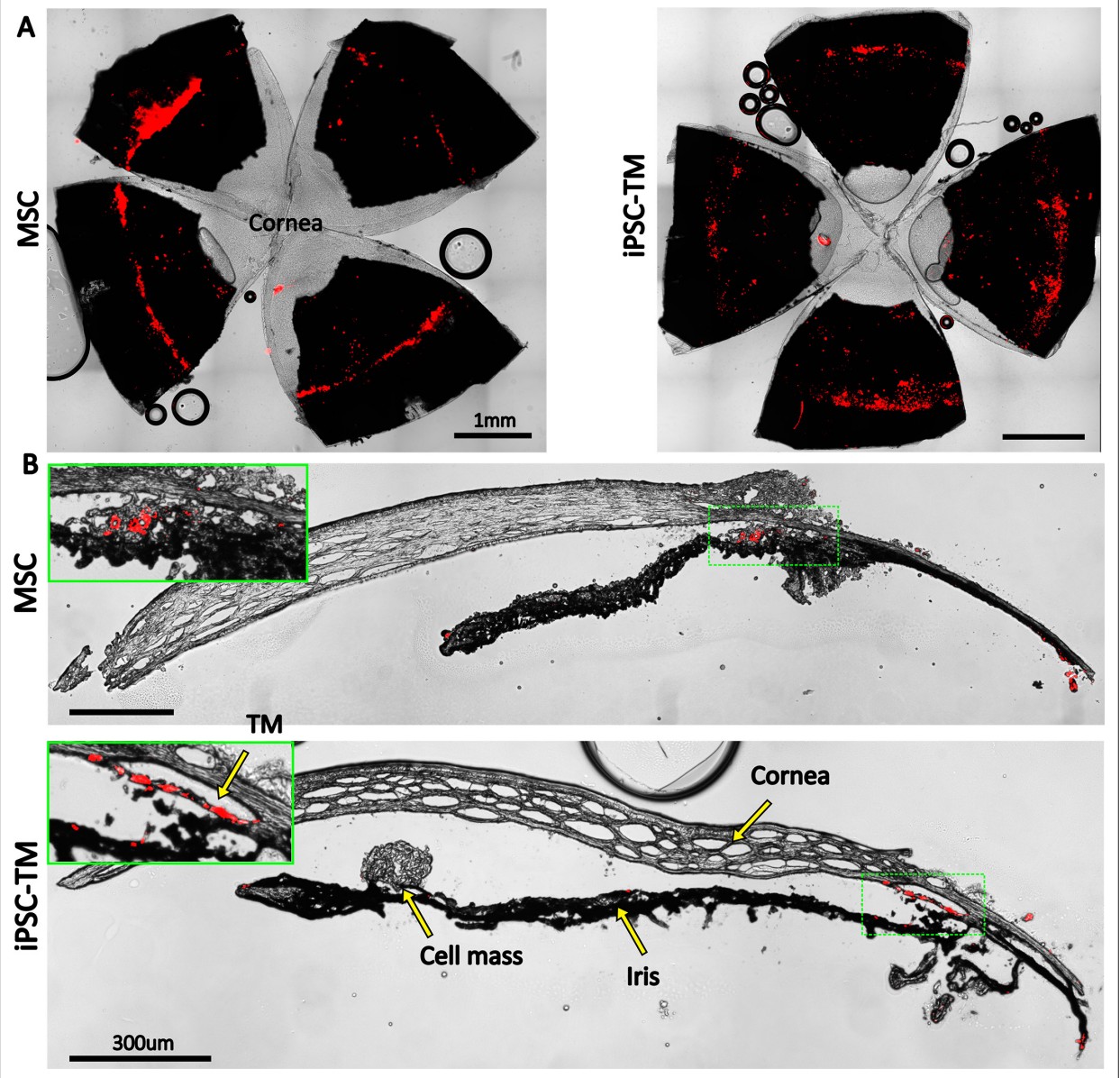

**Figure 6.** Retention of exogenous cells in the anterior segment 3 weeks after injection. Distribution of both human adipose-derived mesenchymal stem cells (hAMSC) and induced pluripotent cell derivative (iPSC-TM) cells (red) are shown in (**A**) en face images of the anterior segments and (**B**) sagittal sections. In panel (**B**), insets show a magnified view of the sites with the most intense fluorescent signal (green boxes). Autofluorescence can be seen in the posterior chambers, as well as exterior to the corneoscleral shell. A cell mass, possibly a growing tumor, can be seen over the iris in the iPSC-TM-injected eye.

The online version of this article includes the following figure supplement(s) for figure 6:

**Figure supplement 1.** Autofluorescence from various ocular tissues at the same fluorescence settings as used in *Figure 6*.

observations after hAMSC treatment when the ~27 months' average lifespan (*Graber et al., 2013*) of the mouse is taken into account. Further studies of factors controlling TM cellularity after hAMSC delivery are indicated but lie beyond the scope of the current study.

Another feature of POAG is an accumulation of ECM within the juxtacanalicular tissue (*Fuchshofer et al., 2003*). The mechanism(s) underlying this ECM accumulation are not entirely understood. Nevertheless, increased levels of transforming growth factor-β2 (TGF-β2) in the AH of POAG patients (*Tripathi et al., 1994*) and its role in decreasing the activity of matrix metalloproteinases (MMPs) suggest that the abnormal ECM deposits may be due to decreased ECM turnover (*Fuchshofer et al., 2003*; *Tamm and Fuchshofer, 2007*). Thus, after detecting the significant increase in TM cellularity

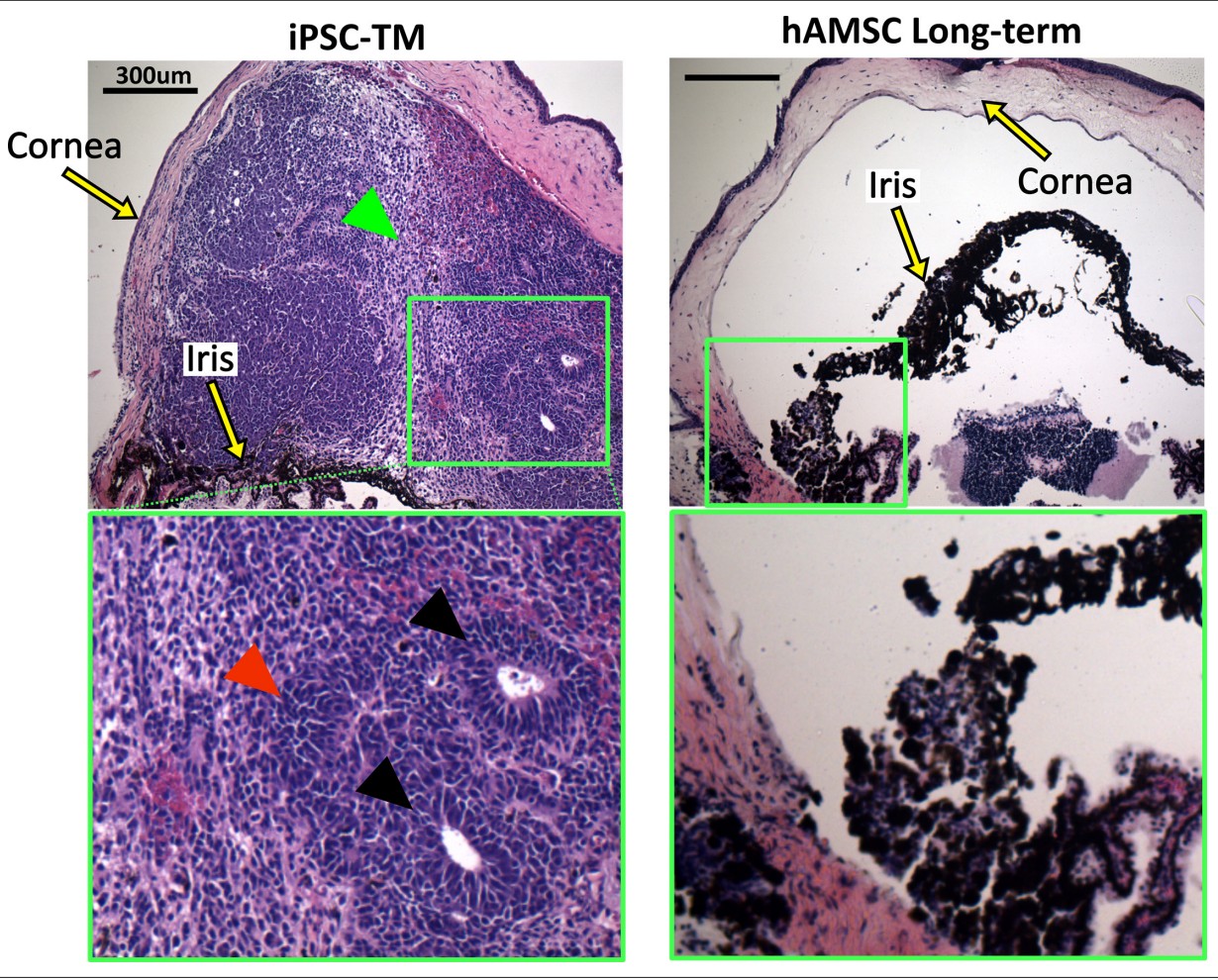

**Figure 7.** Histopathological assessment of tumors in eyes receiving transplanted cells. Induced pluripotent cell derivative (iPSC-TM)- and human adipose-derived mesenchymal stem cell (hAMSC)-transplanted eyes were stained with hematoxylin and eosin (H&E). iPSC-TM sections show distinct tumor characteristics in the anterior chamber, including the presence of rosettes (black arrowheads), densely packed cells with high nuclear-cytoplasmic ratios (red arrowhead), and more loosely coherent cells (green arrowhead). Note that eyes were collected immediately after showing visible signs of tumor growth (usually within a month post-transplantation) and not at a predefined time point. hAMSC eyes at long-term time point showed no sign of tumor growth. In all panels, the green boxes provide a magnified view of the areas where tumor growth or the accumulation of exogenous cells occurred.

and reduction in IOP using hAMSCs, we wondered whether transplanted cells would also affect ECM levels in the TM. Our quantification showed that this was indeed the case: hAMSC-transplanted eyes at the mid-term time point had 35% less BMM under the IW of Schlemm's canal than in saline-injected control eyes. This finding is consistent with the general theme of TM functional restoration seen throughout this study. A future study to analyze the levels of TGF-β2 in the AH, as well as the ratio of active to pro-form levels of MMPs in hAMSC-transplanted eyes, would be of interest to better understand the mechanism through which exogenous cells modulate ECM turnover.

## Comparison with previous work

Unfortunately, it is not feasible to directly compare the results of this study with those of previous studies that have successfully demonstrated the efficacy of nonmagnetic cell therapy in MYOC$^{Y437H}$ mice (*Zhu et al., 2016*; *Zhu et al., 2017*). This is due to the unexpected lack of a POAG phenotype in our transgenic mice (discussed in detail below). Yet in our study, we found a stable IOP lowering and increase in outflow facility over 9 months (corresponding to one-third of the animals' lifespan), which for the first time attests to the possible longevity of IOP lowering due to TM cell therapy. In addition, because of the targeted nature of our delivery technique, we could achieve these reported

therapeutic outcomes by injecting a total of only ~1500 cells, which is significantly lower, i.e., more efficient, than the 50,000 cells used in previous studies (*Zhu et al., 2016*; *Zhu et al., 2017*).

One literature comparison that can be made is for the BMM normalized length. *Li et al., 2021* reported values of 0.40 [0.23, 0.57] for this quantity in naïve eyes and 0.50 [0.38, 0.60] for sham-injected (PBS containing nonconjugated polymeric nanoparticles) eyes in 2- to 3-month-old C57BL/6 mice. *Overby et al., 2014* measured a BMM normalized length of 0.29 [0.16, 0.42] in 6- to 7-month-old mice from the same strain. Despite our sham mid-term group being 9–11 months of age and being on a transgenic background, with no phenotypic manifestation, the 0.52 [0.34, 0.70] BMM normalized length we measured for this group is consistent with those previously reported values. The fact that we observed reduced BMM length in cell-treated eyes vs. control values may suggest that a homeostatic balance in cell-treated eyes was tipped toward the presence of less BMM, consistent with hypotensive IOP measurements and greater outflow facility.

## hAMSCs outperformed iPSC-TMs

This study for the first time compared the IOP-lowering performance of hAMSCs vs. iPSC-TMs – two of the most clinically relevant cell types for future TM cell therapy (*Coulon et al., 2022*). Surprisingly, we found that the performance of iPSC-TMs was significantly inferior to that of hAMSCs, as quantified by several outcome measures; most notably, the IOP reduction after iPSC-TM cell delivery was only half that seen after hAMSC delivery. The beneficial effect of iPSC-TM treatment on TM cellularity was also significantly lower than hAMSC at the short-term time point (1.4-fold vs. 2.2-fold increase), although this difference declined at the mid-term time point (1.4-fold vs. 1.6-fold increase). In addition to their IOP-reducing efficacy, another major drawback of the iPSC-TMs was the high incidence of ocular tumorigenicity. More than 60% of the eyes injected with iPSC-TM cells developed tumors, requiring termination of the experiment. A body of previous literature, including a systematic review of 1000 clinical trials involving mesenchymal stem cell transplantation, finds no incidence of tumorigenicity in tissues receiving mesenchymal stem cells, suggesting an intrinsic resistance to tumor formation once positioned in the correct niche (*Lalu et al., 2012*; *Rodríguez-Fuentes et al., 2021*). On the other hand, tumorigenicity remains a concern for iPSC derivatives due to transfection with oncogenic factors, genetic aberrations during in vitro cultures, and contamination of transplants with undifferentiated cells (*Neri, 2019*; *Lamm et al., 2016*; *Yamanaka, 2020*). Despite following a protocol to isolate differentiated iPSC-TM cells, including using a non-integrating viral vector for transfection of reprogramming factors and a commonly used magnetic activated cell sorting approach (*Yamanaka, 2020*), there unfortunately remains a chance for contamination and reprogramming of these cells posttransplantation. Interestingly, the iris is reported to be a favorable location for organ culture and tumor formation, with fivefold faster growth compared to subcutaneous injection, and thus has previously been considered for tumorigenicity safety studies, emphasizing the importance of rigorous iPSC processing in any future treatments involving iPSC-TM cell injection into the anterior chamber (*Olson and Seiger, 1972*; *Inagaki et al., 2022*; *Boone and DuPree, 1968*).

## Cell retention profiles differed between the two cell types

Both cell types were detectable in the anterior chamber 3 weeks after injection (*Figure 6B*), with iPSC-TM cells tending to better integrate with the TM tissue, whereas the hAMSCs mostly accumulated close to, but not within, the TM. This phenomenon, which was consistently observed, may be due to the widely reported aggregation of mesenchymal stem cells immediately posttransplantation (*Burand et al., 2020*), which consequently prevented them from entering the deeper aspects of the TM, characterized by narrow flow channels. Note that the exogenous iPSC-TMs more directly contributed to increasing TM cellularity than did the hAMSCs due to the better integration of iPSC-TM cells into the TM. This finding complicates the interpretation of the relationship between increased TM cellularity and IOP reduction. In addition, loss of signal in long-term in vivo fluorescent cell tracking is inevitable (*Progatzky et al., 2013*), so the fluorescent signal in *Figure 6* may not be marking all the exogenous cells retained in the anterior eye.

## A likely role for paracrine signaling underlying TM cell therapy

The lack of specific hAMSC homing into the TM also provides important insight about the putative mechanism(s) by which these cells lowered IOP and improved AH dynamics. Several hypotheses

address how injected cells may affect TM functional restoration: exogenous cells can either integrate with the TM and differentiate into TM-like cells, or can promote endogenous TM cell proliferation through direct contact or through their secretome. Du and colleagues, in two studies using mice, showed that TMSCs that reach the TM co-express AQP1 and CHI3L1, indicative of their differentiation into TM cells, although quantification was not performed (*Yun et al., 2018*; *Xiong et al., 2021*). *Zhu et al., 2016* reported a 114% increase in TM cellularity after iPSC-TM injection in MYOC$^{Y437H}$ mice compared to saline-injected controls, yet TM-residing exogenous cells accounted for only 23% of this increase. This finding is consistent with several studies that report the proliferative effect of exogenous cells on the TM in terms of higher Ki-67 expression and BrdU signal, as well as an increased prevalence of Nestin+ progenitor cells (*Manuguerra-Gagné et al., 2013*; *Zhu et al., 2016*; *Zhu et al., 2020*). How this proliferation is mediated, however, is a matter of controversy. In two studies, Zhu et al. showed that iPSC-TMs induce significant proliferation of both cells from the TM5 cell line (an immortalized TM cell line) or primary TM cells carrying Ad5RSV-myocilinY437H when in co-culture, yet did not when they were co-cultured in the presence of a physical (membrane) separation between iPSC-TMs and TM cells (*Zhu et al., 2016*; *Zhu et al., 2020*). Interestingly, *Xiong et al., 2021* conducted similar experiments with TMSCs and MyocY437H primary TM cells and observed no proliferative effect with or without contact between the cells. On the contrary, two studies have reported the beneficial effect of injecting conditioned media from bone marrow MSCs in hypertensive rat eyes, including a significant reduction in IOP, neuroprotection, and elevated proliferation markers in the TM (*Manuguerra-Gagné et al., 2013*; *Roubeix et al., 2015*). In our study, the significant lowering of IOP seen after the delivery of hAMSCs and their accumulation near, but not within, the TM supports the notion that injected cells act upon the TM through their secretome. Thus, a proteomic comparison of the secretome of hAMSCs and iPSC-TMs may provide significant insight into their paracrine effect on TM functional restoration.

## Limitations

The main limitation of this study was the lack of a POAG phenotype in our transgenic mouse colony. Even though MYOC$^{Y437H}$ mice have previously been shown to exhibit an elevation in IOP and a decrease in both outflow facility and TM cellularity, our colony did not show any difference in those parameters compared to WT animals. While we are not certain of the cause, one possibility is that our IOP measurements were inaccurate. However, this is unlikely because of the correlation between measured IOP and outflow facility in cohorts (*Figure 3C*) and the fact that IOPs in our WT animals lay within previously reported ranges (*Zhu et al., 2016*; *Zhu et al., 2017*; *Overby et al., 2014*; *Yun et al., 2014*). It should be noted that reported IOPs for anesthetized WT C57BL/6 mice vary depending on the measurement method used, with a lower bound of ~12.5 mmHg under deep levels of injection-induced anesthesia (*John et al., 1997*) and an upper bound of ~20 mmHg under gas-induced extra light anesthesia (*Li et al., 2021*), with values close to our measurements being frequent in the literature. An alternative, and perhaps more likely, possibility is that the transgene was silenced in the original breeders of this colony. Unfortunately, this only became evident after the 6–7 months' wait time required for the expected onset of the phenotype. Despite the strong effectiveness of our novel TM cell therapy technique (even in the absence of ocular hypertension), the main concern is whether cell therapy would work as effectively in a glaucomatous eye. *Raghunathan et al., 2018* showed that when healthy TM cells are cultured on the ECM derived from glaucomatous TM cells, the healthy cells experience differential stiffening and altered expression profiles similar to the glaucomatous phenotype. Therefore, a glaucomatous ECM may negatively impact the exogenous cells and curtail their therapeutic potential. Fortunately, since in our study, hAMSCs did not seem to need to integrate into the TM to lower IOP, they may also not be affected by glaucomatous changes in the TM. Additionally, Goldmann's equation (*Equation 2*) shows that the same percentage increase in outflow facility produces a greater magnitude of IOP lowering in a hypertensive eye vs. in a normotensive eye. Therefore, evaluation of magnetically steered hAMSC cell therapy in an alternative preclinical glaucoma model is indicated.

An additional limitation is that, since histology was only performed on a subset of the eyes after outflow facility measurements, it is possible that there may also have been undetected tumors in the iPSC-injected eyes reported in the IOP and outflow facility plots. This could affect the reported efficacy of the iPSC-TM cells, and further experiments comparing hADMSCs to more carefully processed iPSC-TM cells may be worthwhile.

In summary, this work shows the effectiveness of our novel magnetic TM cell therapy approach for long-term IOP reduction through functional changes in the conventional outflow pathway. The comparison between hAMSCs and iPSC-TM cells strongly suggested the inferiority of the latter cell type in this treatment paradigm, as judged by tumorigenicity and less effective IOP lowering. The localization of injected hAMSCs deep in the iridocorneal angle, but not full integration into the TM, supports the hypothesis that exogenous cells promote TM functional restoration through paracrine signaling. Therefore, even though the mouse model used in this study did not show a POAG phenotype, this treatment approach merits further study with the eventual goal of clinical translation.

## Materials and methods
### Experimental design
We conducted experiments in several cohorts of mice, as follows:

- WT: wild-type hybrid mice (naïve controls)
- Tg: Tg-MYOCY437H mice, a model of POAG (see details below)
- Sham: Tg mice receiving PBS (injection controls)
- hAMSC: Tg mice receiving magnetically steered hAMSCs
- iPSC-TM: Tg mice receiving magnetically steered iPSC-TMs

Our key outcome measures were IOP, outflow facility, TM cellularity, cell retention in the anterior segment, and ultrastructural analysis of TM ECM, with timelines as indicated in *Figure 1B*. All measurements were made in ex vivo eyes, except for IOP, which was measured longitudinally in living mice.

After breeding and genotyping, mice, regardless of sex, were maintained to age 6–7 months, when transgenic animals were expected to have developed a POAG phenotype. We then made baseline measurements and performed stem cell (or sham) injections and followed animals for various durations:

- Short-term: 3–4 weeks after cell injection
- Mid-term: 3–4 months after cell injection, and
- Long-term: 9 months after cell injection.

Exogenous cell retention in the anterior chamber was measured at only the short-term time point. This is because, in our experience (data not shown), the tracer's signal was only faintly present 2 months after injection in vivo, while signal was maintained for a longer period in vitro, as advertised by the manufacturer. We are unsure whether this loss of signal was caused by a loss of cell integrity or by fluorescence fading in vivo. Further, due to the high incidence of tumorigenicity and inferior overall effectiveness in animals receiving iPSC-TM cells, long-term measurements, as well as ultrastructural analysis, were not pursued for this group. We chose to perform ultrastructural analysis for the hAMSC group at the mid-term time point, as this is the longest time point previously studied (*Zhu et al., 2017*) and enables comparison with previous work.

### Cell preparation
hAMSCs were purchased commercially (Lonza Bioscience, Walkersville, MD, USA) and were prepared for injection as described previously (*Bahrani Fard et al., 2023*). The cells were maintained at 37°C and 5% $CO_2$ in α-MEM supplemented by 10% FBS and 1% penicillin and streptomycin and 2 mM L-glutamine. Cells were passaged using 0.05% trypsin (25-053 CI, Corning Inc, Corning, NY, USA) to detach cells, followed by resuspension and seeding at 5000 cells/cm$^2$ in T-25 cell culture flasks. hAMSCs at 80% confluence (passage 5 or 6) were magnetically labeled by overnight incubation with 150 nm amine-coated SPIONs (SA0150, Ocean NanoTech, San Diego, CA, USA) at 25 µg/ml, followed by inspection under light microscopy to verify sufficient SPION endocytosis. Cells were then trypsinized, resuspended by the addition of cell culture media, and placed in a 1.5 ml microtube. To remove insufficiently magnetized cells, a 0.25″ cubic N52 neodymium magnet was placed on the side of the tube, resulting in rapid formation of a cell pellet close to the magnet. Supernatant and nonmagnetic cells were then removed.

In exogenous cell retention studies, the cells remaining in the microtube were labeled using the PKH26 lipophilic dye kit (Sigma-Aldrich, St. Louis, MO, USA) according to the manufacturer's

instructions. In brief, a cell solution was prepared in the diluent component of the kit and was vigorously mixed with an equal volume of the 4 μM dye solution. After 3 min at room temperature, an equal volume of FBS was added to the cell solution to stop the reaction, and cells were washed three times with the cell culture media to remove any unbound dye. For all the experiments where animals received hAMSC, cell count and >90% viability were verified using a Countess II Automated Cell Counter (Thermo Fisher Scientific, Waltham, MA, USA). The cells were then resuspended in sterile PBS (1X, Thermo Fisher Scientific, Waltham, MA) to a final concentration of 1k cells/μl.

Mouse iPSC-TMs have previously been developed and characterized (*Zhu et al., 2016*). In brief, mouse dermal fibroblasts are reprogrammed through Sendai virus-mediated reprogramming with the transcription factors OCT4, SOX2, KLF4, and c-MYC. The pluripotency of reprogrammed iPSCs was confirmed using RT-PCR, immunocytochemistry, immunoblotting, and teratoma formation. iPSCs were then differentiated by culturing in conditioned media from primary human TM (phTM) cells. To prepare this conditioned media, phTM cells were extracted from donor eyes and cultured in α-MEM supplemented by 10% inactivated FBS and 2% primocin. Conditioned media was then collected from the cells and sterilized by passing through a 0.2 μm membrane filter. The iPSCs were maintained in conditioned media for 8 weeks to induce differentiation. It is important to remove any undifferentiated iPSCs from the iPSC-TM populations due to the risk of tumorigenicity associated with pluripotent stem cells. Therefore, the iPSC-TMs were incubated with CD15 antibodies (Miltneyi Biotec, Bergisch Gladbach, Germany) conjugated with magnetic microbeads to label the undifferentiated iPSCs. Then, the cells were washed, loaded into a MACS LD column, and were placed in a magnetic separator (Miltenyi Biotec, Germany).

## Transgenic mice

All animal procedures were approved by the Georgia Tech Institutional Animal Care and Use Committee (approved IACUC protocol number A100282) and performed in conformance with the ARVO Statement for the Use of Animals in Ophthalmic and Vision Research. Breeder pairs of C57BL/6 Tg-MYOCY437H mice were shipped from Iowa to a quarantining facility (Charles River, Wilmington, MA, USA), underwent IVF rederivation, and were shipped to Georgia Tech after ~4 months. Breeders carrying one copy of the transgene on a C57BL/6 background were crossed with SJL mice (Charles River) of similar age, with half of the hybrid offspring carrying the transgene. Pups were genotyped using human *MYOC* primers (forward: CGTGCCTAATGGGAGGTCTAT; reverse: CTGGTCCAAGGT CAATTGGT). Only F1 animals were used in studies.

## Cell injections

Cell injection needles were fabricated as described previously (*Bahrani Fard et al., 2023*). In brief, glass micropipettes were pulled using a pipette puller (P-97, Sutter Instruments, Novato, CA, USA), and the tips were broken and beveled at a 30° on a microelectrode beveler (BV-10, Sutter Instruments, Novato, CA, USA), followed by rotating to both sides for enhanced sharpness (tri-beveling). The resulting needle had a pointed tip and an outer diameter of approximately 100 μm. Cell adhesion to the needle walls in the lumen can cause inconsistent cell delivery to the eye; thus, we plasma-cleaned the needles, coated them with trichlorosilane, and loaded them with 0.02% Pluronic F-127 (P2443, Sigma-Aldrich) for 1 hr at room temperature, followed by vigorously rinsing with PBS. Needles were sterilized with 70% ethanol prior to injections.

Each animal was prepared for unilateral injection of cells by applying a tropicamide eyedrop (Bausch & Lomb, Bridgewater, NJ, USA) to start pupil dilation before inducing anesthesia using an induction chamber receiving 2.5% isoflurane at 600 ml/min. Once the toe-pinch reflex was lost, the animal was transferred to a heated bed, and the head was immobilized with Velcro straps while anesthesia was maintained through a nose cone. A drop of tetracaine (Bausch & Lomb) was applied to the eye being injected while the contralateral eye received ophthalmic lubricant (Systane Ultra, Alcon, Geneva, Switzerland) to prevent drying. The needle, mounted on an injector assembly (MMP-KIT, WPI, Sarasota, FL, USA), was attached to a micromanipulator and connected to a microsyringe pump (PHD Ultra, Harvard Apparatus, Holliston, MA, USA). The needle was filled with 3 μl of the injection solution (either cells or PBS for control [sham] injections), aligned at a 30° angle with the eye, and advanced into the AC in a swift motion until the tip was located approximately in the center of the AC while the eye was held in a proptosed position using a pair of nonmagnetic forceps (*Figure 1A*).

A total of 1.5 µl of the solution was injected at 2.4 µl/min and if the solution contained cells, a point magnet (a thin stainless steel rod attached to a permanent magnet) was used to magnetically steer the cells toward the TM in a continuous motion for the duration of cell ejection from the needle, as previously described in detail (*Bahrani Fard et al., 2023*). Injected eyes received ophthalmic antibiotic combination ointment (neomycin, polymyxin, bacitracin) and were kept on a heated bed until recovery from anesthesia.

### IOP measurements

We measured the IOPs between 1 and 3 pm (to minimize diurnal variations) by first placing the mouse in an induction chamber until the righting reflex was lost and breathing slowed. The animal was then transferred to a heated platform, secured with straps, and a tonometer (TonoLab, iCare, Vantaa, Finland) mounted on a micromanipulator (M3301, WPI, Sarasota, FL, USA) was aligned perpendicular to the corneal surface at the center of the cornea. Eight IOP measurements were taken from each eye, and the reported IOP was the average of all eight measurements. Even though some labs exclude the highest and lowest of the eight measurements from the IOP average (*McDowell et al., 2022*), we did not observe significant intra-measurement variability and thus included all the eight replicates when determining the IOP. The entire duration of IOP measurement was typically 3 min or less, which is less than has been reported for the start of significant anesthesia-induced IOP reduction (*Qiu et al., 2014*; *Tsuchiya et al., 2021*).

### Measurement of outflow facility

Outflow facility, which quantifies the ease of fluid drainage from the eye, is defined as the ratio of steady-state outflow rate over IOP in an enucleated eye. We measured facility in enucleated eyes using the previously established iPerfusion system (*Sherwood et al., 2016*). The system's sensors were calibrated before each measurement session to ensure reliability and the absence of bubbles or leaks, which can cause large errors in the measurements. Animals were euthanized by intraperitoneal injection of sodium pentobarbital, and eyes were enucleated by sliding a pair of fine-angled forceps behind the eye through the nasal side of the eye socket and pulling the eye out by grabbing onto the optic nerve and the surrounding retrobulbar tissues. The posterior of the eye was secured to a mounting post using a very small drop of cyanoacrylate adhesive (Superglue) inside a heated water bath (35°C) filled with DPBS supplemented with 5.5 mM glucose. A beveled micropipette, mounted on a micromanipulator, was then used to cannulate the eye at a 30° angle. The eyes were stabilized at an IOP of 8 mmHg for 30 min and then perfused at eight evenly distributed pressure steps starting from 4.5 mmHg and finishing at 16.5 mmHg while flow rate and pressure data were acquired (*Figure 3Ai–iii*). The resulting flow-pressure data were fit with an empirical power-law relationship (*Sherwood et al., 2016*)

$$C\left(P\right) = C_r \left(\frac{P}{P_r}\right)^{\beta} \tag{1}$$

where $C$ is the steady-state outflow facility calculated for each pressure step, $P$ is the steady-state pressure for that step, and the subscript $r$ refers to the parameter evaluated at the reference pressure of 8 mmHg which corresponds to the physiologic pressure difference across the conventional outflow pathway (*Sherwood et al., 2016*). $\beta$ is a nonlinearity parameter that is determined by data fitting along with $C_r$. A total of 114 eyes were randomly chosen for outflow facility measurements, of which 9 were excluded due to failed perfusion (e.g. poor cannulation).

### Comparison between experimental and expected IOP

Steady-state AH dynamics can be described by the modified Goldmann equation (*Brubaker, 2004*):

$$Q = Q_{in} - Q_0 = C\left(IOP - P_e\right) \tag{2}$$

where $Q_{in}$ is the rate of AH humor formation, $Q_0$ is the uveoscleral (unconventional) outflow rate, and $P_e$ is the episcleral venous pressure. Since the left-hand side of *Equation 2* is essentially pressure-independent, $Q$ was assumed to be the same across all the experimental groups.

To cross-validate the IOP and outflow facility measurements, we calculated $Q$ in WT animals by inserting the mean measured outflow facility and mean measured IOP from WT animals into *Equation 2*, assuming $P_e$ to be 7 mmHg (*Sherwood et al., 2016*). Average outflow facilities for all the groups were then adjusted using *Equation 1* to account for the pressure dependence of facility. Using these adjusted facilities and assuming $Q$ to be the same for all groups, we calculated an 'expected IOP' for each experimental group, which can be interpreted as the IOP that is consistent with the measured outflow facility.

## Histology, histopathology, and morphometric studies

Similar to the procedure used previously (*Bahrani Fard et al., 2023*), all experimental eyes were immersion fixed in 10% formalin (Fisher Healthcare, Waltham, MA, USA) overnight at 4°C after the corresponding in vivo and ex vivo measurements (no measurements were performed on the eyes used for exogenous cell retention study). Of these eyes, a total of 59 were randomly selected from various groups for TM cellularity quantifications. Eyes were then dissected under a surgical microscope, and isolated anterior segments were cut into four leaflets. This anterior segment whole mount was placed on a glass slide with the cornea facing up and mounted with PBS. A Leica DMB6 epifluorescent microscope (Leica Microsystems, Wetzlar, Germany) was used to create fluorescent en face tilescan images. Two quadrants of each whole mount were then prepared for cryosectioning. These quadrants received sequential 15 min treatments in 15% sucrose (Sigma-Aldrich, St. Louis, MO, USA), 30% sucrose, and a 1:1 solution of 30% sucrose and optimal cutting temperature (OCT) media. After embedding in OCT, samples were floated in a 100% ethanol bath cooled by dry ice to flash-freeze. 10-µm-thick sagittal sections were cut using a CryoStar NX70 cryostat (Thermo Fisher Scientific, Waltham, MA, USA) and placed on Superfrost Plus gold slides (Thermo Fisher Scientific, Waltham, MA, USA). In an additional step, specific to TM cellularity quantification, the samples were permeabilized with 0.2% Triton X-100 for 10 min and stained with DAPI (NucBlue Fixed Cell DAPI, Invitrogen, Waltham, MA, USA) for 15 min followed by coverslipping with antifade media (ProLong Gold Antifade medium, Invitrogen, Waltham, MA, USA). Sagittal sections were then imaged as tilescans.

To quantify TM cellularity, ideally all the cells in the TM should be counted. However, due to the partial or complete collapse of Schlemm's canal and the small separation between the TM and the iris in the murine iridocorneal angle (*Li et al., 2019*), identifying the boundaries of the TM can be challenging. Thus, to minimize error, we instead counted the DAPI-stained nuclei in the region that we could identify as the TM with a high confidence and normalized this count by the length of the IW of Schlemm's canal adjacent to this segment (*Figure 4A* and *Figure 4—figure supplement 1*). Note that morphological characteristics such as the autofluorescence in the corneoscleral shell, high degree of pigmentation in the iris, as well as the change in the density and orientation of the cells transitioning from the TM to iris helped with locating the TM cell nuclei.

Eyes of animals injected with iPSC-TMs showing anatomical signs of tumor growth were enucleated and immersion fixed in 10% formalin for histopathological studies. Three of these eyes were randomly chosen, dehydrated, and embedded in paraffin. Subsequently, 5-µm-thick sagittal sections were cut using a microtome (Thermo Fisher Scientific, Waltham, MA, USA) and stained with hematoxylin and eosin (H&E). Three additional eyes from the hAMSC long-term group underwent the same procedure for comparison.

## Quantification of ECM underlying the IW of SC

The amount of BMM in the hAMSC mid-term experimental group and in corresponding injection control eyes (four eyes in each cohort) was quantified using electron micrographs, using an approach similar to that previously described (*Li et al., 2021*; *Overby et al., 2014*). In brief, the two anterior segment quadrants not used for TM cell counting (described above) were immersion fixed overnight at 4°C in universal fixative (2.5% glutaraldehyde, 2.5% paraformaldehyde in Sörensen's buffer). The specimens were next embedded in Epon resin, and 65 nm sagittal sections were cut through iridocorneal tissues using an ultramicrotome (Leica EM UC6, A-1170; Leica Mikrosysteme GmbH) followed by staining with uranyl acetate/lead citrate. Sagittal sections were examined with a JEM-1400 electron microscope (JEOL USA, Peabody, MA, USA) at ×8000 magnification. At least one section per quadrant was included in the quantification of BMM deposits as described below.

The lengths of BMM segments directly in contact with the IW of Schlemm's canal and the total length of IW were measured from electron micrographs using ImageJ (*Schneider et al., 2012*) by two independent annotators in a masked fashion. To enhance BMM identification from micrographs, we first adjusted the contrast for each image, using an ImageJ macro that assigned a brightness level of 255 to a manually selected region of the lumen of SC and a brightness level of zero to a manually selected region of a cell nucleus. The ratio of the length of BMM segments directly underlying the IW to the total length of IW was calculated for each segment, representing the extent of non-fenestrated BMM material underlying the IW. *Figure 5—figure supplement 1* exhibits an example of the demarcations. Note that ECM deposits clearly separated from the IW were not considered as BMM deposits.

## Statistical analysis

IOP, outflow facility, and TM cellularity index were tested for normality using the Shapiro-Wilk test for each treatment group. Since the outflow facility is known to be log-normally distributed (*Sherwood et al., 2016*), facility data was first log-transformed prior to conducting any statistical tests. All outcome measures, except for TM cellularity and normalized BMM length, were analyzed by one-way ANOVA. For TM cellularity and BMM length, we used a linear mixed-effects model, treating the experimental group as the fixed effect while considering the eyes, various sections of each eye, and annotators (for BMM only) as replicates, i.e., as random effects. Following these analyses, we conducted post hoc comparisons with Bonferroni correction. However, we limited our comparisons to those chosen a priori to be relevant to the interpretation of our study to avoid an overly conservative adjustment of critical p-values as required by Bonferroni correction. To compare the impact of different treatments on IOP and outflow facility, we computed the difference between the treatment groups and their respective injection controls. Subsequently, we conducted two-tailed t-tests with Bonferroni correction. Given the log-transformation of facility data, the subtracted values became ratios upon inverse transformation. To check for consistency between IOP and outflow facility measurements, we calculated residuals as the difference between the expected and experimentally measured IOPs. A two-tailed t-test was then performed on these residuals with $H_0: = 0$. Pearson's correlation test was used to measure the relation between TM cellular density and IOP. All analyses were done using MATLAB (v2020, MathWorks, Natick, MA, USA). A significance level of 5% was employed for all tests, unless explicitly stated otherwise.

## Acknowledgements

National Institutes of Health grant R01 EY030071 (CRE, SYE, MHK); The Georgia Research Alliance (CRE).

## Additional information

### Funding

| Funder | Grant reference number | Author |
| --- | --- | --- |
| National Institutes of Health | R01 EY030071 | Stanislav Y Emelianov<br>Markus H Kuehn<br>C Ross Ethier |
| Georgia Research Alliance | Endowed chair | C Ross Ethier |

The funders had no role in study design, data collection and interpretation, or the decision to submit the work for publication.

### Author contributions

M Reza Bahranifard, Conceptualization, Data curation, Investigation, Visualization, Methodology, Writing - original draft, Writing – review and editing; Jessica Chan, A Thomas Read, Guorong Li, Babak N Safa, Seyed Mohammad Siadat, Anamik Jhunjhunwala, Hans E Grossniklaus, Investigation, Writing – review and editing; Lin Cheng, Investigation, Methodology, Writing – review and editing; Stanislav Y Emelianov, Conceptualization, Supervision, Funding acquisition, Writing – review and editing; W

Daniel Stamer, Conceptualization, Supervision, Project administration, Writing – review and editing; Markus H Kuehn, Conceptualization, Supervision, Funding acquisition, Project administration, Writing – review and editing; C Ross Ethier, Conceptualization, Resources, Supervision, Funding acquisition, Project administration, Writing – review and editing

Author ORCIDs
M Reza Bahranifard ![ORCID] https://orcid.org/0009-0005-1369-8772
A Thomas Read ![ORCID] http://orcid.org/0000-0002-2764-6413
Guorong Li ![ORCID] http://orcid.org/0009-0002-4047-8546
Babak N Safa ![ORCID] http://orcid.org/0000-0002-1849-2870
Seyed Mohammad Siadat ![ORCID] http://orcid.org/0000-0002-5063-0305
Anamik Jhunjhunwala ![ORCID] https://orcid.org/0000-0002-7126-4463
W Daniel Stamer ![ORCID] http://orcid.org/0000-0002-2504-8997
C Ross Ethier ![ORCID] https://orcid.org/0000-0001-6110-3052

## Ethics

This study was performed in strict accordance with the recommendations in the Guide for the Care and Use of Laboratory Animals of the National Institutes of Health. All animal procedures were approved by the Georgia Tech Institutional Animal Care and Use Committee (approved protocol A100282) and performed in conformance with the ARVO Statement for the Use of Animals in Ophthalmic and Vision Research.

Reviewer #1 (Public review): https://doi.org/10.7554/eLife.103256.3.sa1
Reviewer #2 (Public review): https://doi.org/10.7554/eLife.103256.3.sa2
Author response https://doi.org/10.7554/eLife.103256.3.sa3

# Additional files

## Supplementary files

MDAR checklist

Source data 1. Summary of IOP, Outflow Facility, TM Cellularity, Expected IOP (computed from facility), and extracellular matrix (ECM) length under the inner wall.

## Data availability

All data generated or analyzed during this study are included in the manuscript and supporting files; source data files have been provided as a supplementary attachment (*Source data 1*).

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

## Appendix 1

### SPION visualization posttransplantation

For reasons described below, it was useful to know where SPIONs had accumulated within the eye. We therefore carried out histological analyses of tissue sections to visualize SPIONs, using a Prussian blue staining process.

### Prussian blue staining of SPIONs

A Prussian blue solution was freshly prepared by mixing a 1:1 ratio of a 20% aqueous solution of hydrochloric acid and a 10% aqueous solution of potassium ferrocyanide ($K_4Fe(CN)_6 \cdot 3H_2O$, Sigma-Aldrich). Cryosections from the eyes, which were sampled for cell retention studies, were rehydrated for 30 min. Subsequently, the sections were covered with the 10% potassium ferrocyanide solution for 5 min, followed by a 15 min treatment with the Prussian blue solution. After treatment, the sections were rinsed three times in distilled water, dehydrated through increasing concentrations of ethanol, cleared in xylene, and then mounted for imaging.

### SPIONs co-located with exogenous cells posttransplantation

A major concern regarding the use of SPIONs for cell encapsulation is dose-dependent toxicity, which could damage both the transplanted cells and native tissues (*Sadeghiani et al., 2005*; *Stroh et al., 2004*). We therefore visualized the SPIONs using Prussian blue staining anterior segment sagittal sections (*Appendix 1—figure 1*). Unfortunately, this dark blue stain proved barely discernible from pigment. Detectable labeling could mostly be found at comparable locations within the AC as in hAMSC-injected eyes (*Figure 6B*), where both the cells and SPIONs accumulated in the vicinity of TM, and in iPSC-TM-injected eyes (*Figure 6B*), with the cells and SPIONs found within the TM. Prussian blue did not stain materials in the saline-injected control eyes.

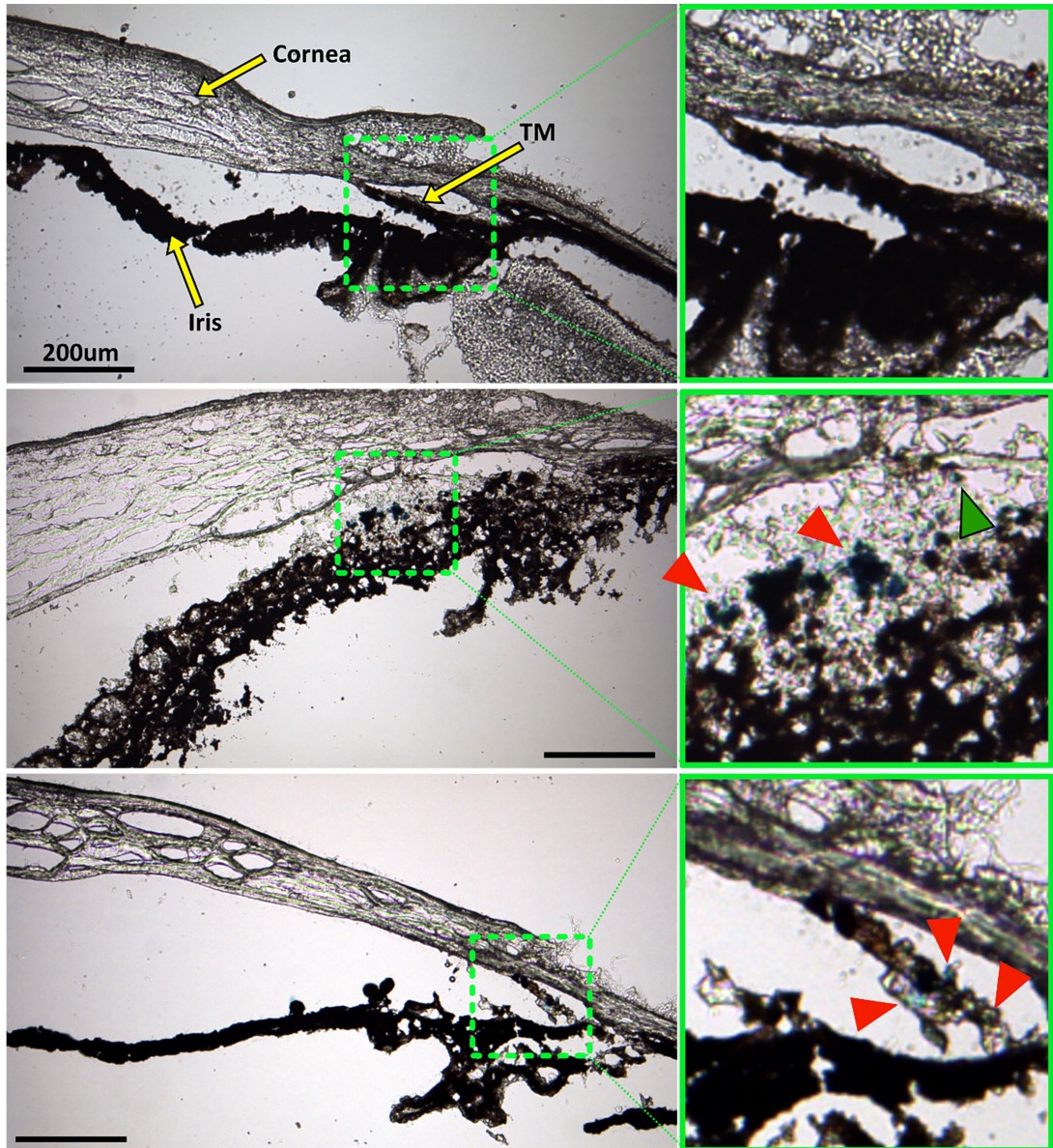

**Appendix 1—figure 1.** Prussian blue staining to locate superparamagnetic iron oxide nanoparticles (SPIONs) within the anterior segment after cell transplantation. The left column shows overview images of the anterior segment, while green boxes in the right column show a zoomed view of the region with strongest Prussian blue staining, corresponding to the green dashed boxes in the left column. Top row: No Prussian blue staining could be found in the saline injection control. Middle row: Prussian blue stain is challenging to distinguish from melanin, but accumulation of blue label (red arrowheads) can be seen to coincide with the locations of exogenous cells visualized in *Figure 6*. In particular, injected human adipose-derived mesenchymal stem cells (hAMSCs) primarily accumulated close to the trabecular meshwork (TM), corresponding to the location of Prussian blue stain. A small region of Prussian blue staining can be observed in the TM (green arrowhead). Bottom row: Similarly, in eyes receiving induced pluripotent cell derivatives (iPSC-TMs), most of the Prussian blue staining was found within the TM, corresponding to the location of injected cells (*Figure 6*). Unfortunately, the fluorescent signal in *Figure 6* was significantly attenuated after Prussian blue staining and could not be overlaid on these images to assist with interpretation. Iris degradation, notable in the middle row, is an undesirable artifact of the cryosectioning or staining process (*Figure 6*).

## SPIONs did not accumulate within the native tissues of the AC

Iron oxide-induced toxicity, as both cytotoxicity and genotoxicity, is a major concern when using SPIONs in cell encapsulation and transplantation. In vitro studies have reported that SPION labeling

is generally safe at concentrations below 100 μg/ml (*Singh et al., 2010*). Since we used a fourfold lower concentration for cell labeling in this study, the encapsulated cells were likely unaffected.

Once inside the AC, the SPIONs may be released from the injected cells. The TM, as the phagocytic and filtering component of the main AH outflow pathway, is a likely destination. Our Prussian blue staining to visualize SPIONs in the AC after delivery was masked by pigmentation and was hard to visualize (*Appendix 1—figure 1*). The SPIONs that we could detect were mostly co-located with the transplanted cells and were likely not released at a significant rate within the AC. In the case of iPSC-TM cells, which showed good integration with the TM, detectable SPIONs were also primarily found within the TM. Whether these SPIONs had been released from the injected cells or were still encapsulated remains unknown. Nevertheless, the significant increase in TM cellularity discussed above indicates that the accumulation of SPIONs within the TM is unlikely to have any toxic effect on native tissues.

# Appendix 2

## Detailed analysis of IOP measurements

**Appendix 2—table 1.** Intraocular pressure (IOP) measurements for select groups, shown as means and [95% confidence intervals] similar to Table 1.
Tg Mid: Untreated eyes, either contralateral to sham mid-term or human adipose-derived mesenchymal stem cell (hAMSC) mid-term eyes (n=23).

| Group | IOP (mmHg) |
|---|---|
| WT | 15.6 [14.8,16.3] |
| Tg | 15.4 [14.7,16.1] |
| Tg Mid | 16.5 [15.9,17.1] |
| Sham Mid | 16.9 [15.5,18.2] |

**Appendix 2—table 2.** Result of multiple comparison for the groups listed in Appendix 2—table 1. None of the comparisons reached statistical significance.

| | Compared groups | | |
|---|---|---|---|
| | vs. | | p-Value* |
| 1 | WT | Tg | 0.3579 |
| 2 | WT | Tg Mid | 0.0680 |
| 3 | WT | Sham Mid | 0.0491 |
| 4 | Tg | Tg Mid | 0.0278 |
| 5 | Tg | Sham Mid | 0.0311 |
| 6 | Tg Mid | Sham Mid | 0.5148 |

*Post hoc comparisons were performed after ANOVA. Bonferroni correction was used to adjust the critical p-value from 0.05 to 0.008 (based on the six reported comparisons).

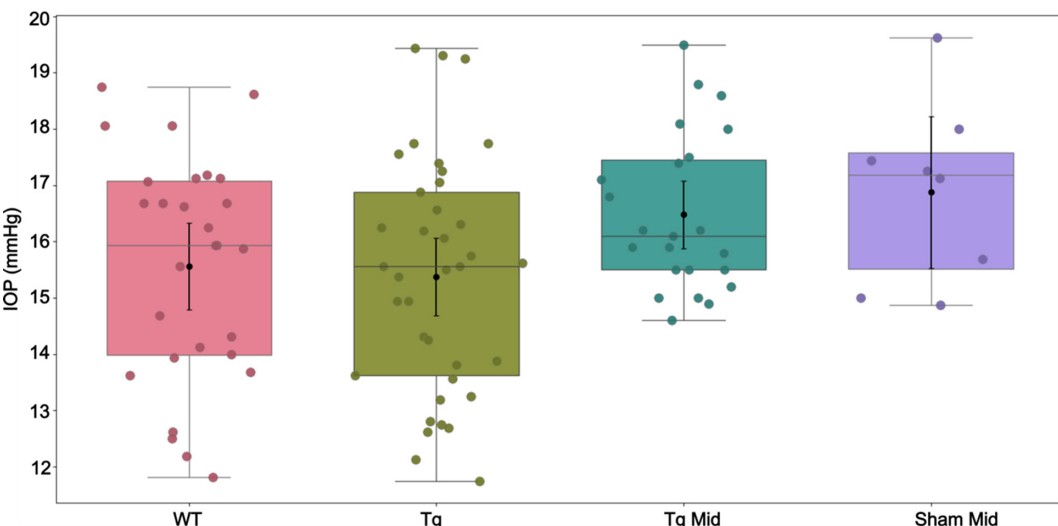

**Appendix 2—figure 1.** Intraocular pressure (IOP) measurements corresponding to *Appendix 2—table 1*. In each experimental cohort, the box plot shows interquartile range, and the center horizontal line denotes median. Black dots and their whiskers show the mean and 95% confidence interval on the mean, respectively. Dots represent individual eyes. See *Appendix 2—table 2* for complete statistical analysis.

