## [Editor Report · eLife Assessment]

This study has **fundamental** findings that support the potential application of exogenous stem cell therapy as a viable therapeutic option for the management of intraocular pressure (IOP) and to increase outflow facility. The evidence supporting the clinical application of stem cells is **compelling**, using a combination of established in vivo and ex vivo experimental techniques. The work will be of interest to both basic stem cell biologists and clinical glaucoma specialists.

---

## [Referee Report · Reviewer #1 (Public review)]

Summary:

This manuscript describes a novel magnetic steering technique to target human adipose derived mesenchymal stem cells (hAMSC) or induce pluripotent stem cells to the TM (iPSC-TM). The authors show delivery of the stem cells lowered IOP, increased ouflow facility, and increased TM cellularity.

Strengths:

The technique is novel and shows promise as a novel therapeutic to lower in IOP in glaucoma. hAMSC are able to lower IOP below baseline as well as increase outflow facility above baseline with no tumorigenicity. These data will have a positive impact on the field and will guide further research using hAMSC in glaucoma models.

Weaknesses:

The transgenic mouse model of glaucoma the authors used did not show ocular hypertensive phenotypes as previously reported; therefore, the Tg-MYOCY437H model should be used with caution in the future. However, the results presented here clearly show magnetically steered cell therapy as a viable treatment strategy to lower intraocular pressure even from baseline. Future studies are needed to demonstrate the effects in ocular hypertensive eyes.

---

## [Referee Report · Reviewer #2 (Public review)]

This observational study investigates the efficacy of intracameral injected human stems cells as a means to re-functionalize the trabecular meshwork for the restoration of intraocular pressure homeostasis. Using a murine model of glaucoma, human adipose-derived mesenchymal stem cells are shown to be biologically safer and functionally superior at eliciting a sustained reduction in intraocular pressure (IOP). The authors conclude that the use of magnetically-steered human adipose-derived mesenchymal stem cells has potential for long-term treatment of ocular hypertension in glaucoma.

Comments on revisions: Previously noted concerns have been thoughtfully and sincerely considered by the authors and are now clearly addressed in the revised manuscript. No further concerns/comments.

---

## [Author Response]

The following is the authors’ response to the original reviews

**Public Reviews:**

**Reviewer #1 (Public review):**
Summary:This manuscript describes a novel magnetic steering technique to target human adipose derived mesenchymal stem cells (hAMSC) or induce pluripotent stem cells to the TM (iPSC-TM). The authors show that delivery of the stem cells lowered IOP, increased outflow facility, and increased TM cellularity.Strengths:The technique is novel and shows promise as a novel therapeutic to lower IOP in glaucoma. hAMSC are able to lower IOP below the baseline as well as increase outflow facility above baseline with no tumorigenicity. These data will have a positive impact on the field and will guide further research using hAMSC in glaucoma models.Weaknesses:The transgenic mouse model of glaucoma the authors used did not show ocular hypertensive phenotypes at 6-7 months of age as previously reported. Therefore, if there is no pathology in these animals the authors did not show a restoration of function, but rather a decrease in pressure below normal IOP.

We appreciate the reviewer’s feedback and agree with the statement of weakness. Accordingly, we have revised the language to improve clarity. Specifically, all references to "restoration of IOP" or "restoration of conventional outflow function" have been replaced with more precise phrases, in the following locations:

• lines 2-3 (title): Magnetically steered cell therapy for reduction of intraocular pressure as a treatment strategy for open-angle glaucoma

• lines 36-8 (abstract): We observed a 4.5 [3.1, 6.0] mmHg or 27% reduction in intraocular pressure (IOP) for nine months after a single dose of only 1500 magnetically-steered hAMSCs, explained by increased conventional outflow facility and associated with higher TM cellularity.

• lines 45-6 (one-sentence summary): A novel magnetic cell therapy provided effective intraocular pressure reduction in mice, motivating future translational studies.

• lines 123-4 (introduction): Despite the absence of ocular hypertension in our MYOC^Y437H^ mice, our data demonstrate sustained IOP lowering and a significant benefit of magnetic cell steering in the eye, particularly for hAMSCs, strongly indicating further translational potential.

• line 207 (results): The observed reductions in IOP and increases in outflow facility after delivery of both cell types suggested functional changes in the conventional outflow pathway.

• line 509-10 (discussion): In summary, this work shows the effectiveness of our novel magnetic TM cell therapy approach for long-term IOP reduction through functional changes in the conventional outflow pathway.

It is very important to note that at the 23rd annual Trabecular Meshwork Study Club meeting (San Diego, December 2024), Dr. Zode, the lead author of reference 26 originally describing the transgenic myocilin mouse model, announced during his talk that this model no longer demonstrates the glaucomatous phenotype in his hands, which incidentally has motivated him to create a new, CRISPR MYOC mouse model. Dr. Zode also stated that he was uncertain of the reason for this loss of phenotype. His observation is consistent with our report. However, other investigators continue to observe the desired phenotype in their colonies of this mouse (Dr. Wei Zhu, personal communication). Continued use of this mouse model should therefore be approached with caution.

**Reviewer #2 (Public review):**
Summary:This observational study investigates the efficacy of intracameral injected human stem cells as a means to re-functionalize the trabecular meshwork for the restoration of intraocular pressure homeostasis. Using a murine model of glaucoma, human adiposederived mesenchymal stem cells are shown to be biologically safer and functionally superior at eliciting a sustained reduction in intraocular pressure (IOP). The authors conclude that the use of human adipose-derived mesenchymal stem cells has the potential for long-term treatment of ocular hypertension in glaucoma.Strengths:A noted strength is the use of a magnetic steering technique to direct injected stem cells to the iridocorneal angle. An additional strength is the comparison of efficacy between two distinct sources of stem cells: human adipose-derived mesenchymal vs. induced pluripotent cell derivatives. Utilizing both in vivo and ex vivo methodology coupled with histological evidence of introduced stem cell localization provides a consistent and compelling argument for a sustainable impact exogenous stem cells may have on the refunctionalization of a pathologically compromised TM.Weaknesses:A noted weakness of the study, as pointed out by the authors, includes the unanticipated failure of the genetic model to develop glaucoma-related pathology (elevated IOP, TM cell changes). While this is most unfortunate, it does temper the conclusion that exogenous human adipose derived mesenchymal stem cells may restore TM cell function. Given that TM cell function was not altered in their genetic model, it is difficult to say with any certainty that the introduced stem cells would be capable of restoring pathologically altered TM function. A restoration effect remains to be seen.

We acknowledge that the phrase “restoration of TM function” is not fully supported by our results, given the absence of ocular hypertension in our animal model. Accordingly, we have revised the language to more precisely describe our findings. For specific details regarding these changes, please refer to our response to Reviewer 1’s public comments above.

Another noted complication to these findings is the observation that sham intracameralinjected saline control animals all showed elevated IOP and reduced outflow facility, compared to WT or Tg untreated animals, which allowed for more robust statistically significant outcomes. Additional comments/concerns that the authors may wish to address are elaborated in the Private Review section.

We agree that sham-injected animals tended to have higher average IOPs than transgenic animals in our study. However, these differences did not reach statistical significance and therefore remain inconclusive. Further, an increase in IOP following placebo injection has been previously reported (Zhu et al., 2016).

Prompted by the Referee’s comments and also a private comment from Referee 1, we further investigated this effect by analyzing IOP in uninjected contralateral eyes at the mid-term time point and comparing the IOPs in these eyes to other cohorts, as now presented as additional data in Supplementary Tables 1 and 2 and Supplementary Figure 4 (see below). In brief, the uninjected contralateral transgenic eyes (10 months old) showed an IOP of 16.5 [15.9, 17.1] mmHg, which was intermediate between the IOP levels of the 6–7-month-old Tg group (15.4 [14.7, 16.1] mmHg) and the sham group (16.9 [15.5, 18.2] mmHg). However, none of these differences reached statistical significance. Additionally, we cannot rule out potential contralateral effects induced by the injections.

Regarding the best way to assess the effect of cell treatment, we feel very strongly that the most relevant IOP comparison is between cell-injected eyes and control (vehicle)-injected eyes, since this provides the most direct accounting for the effects of injection itself on IOP. Other comparisons, such as WT or untreated Tg eyes vs. cell-treated eyes, are interesting but harder to interpret. However, in response to the referee’s comment, we have added comparisons between cell-treated groups and untreated Tg eyes to Table 2, adjusting the post-hoc corrections accordingly. All hAMSC treated groups show statistically significant decrease in IOP even compared to Tg untreated eyes, while iPSC-TMs fail to reach such significance.

The following changes were made to the manuscript:

Lines 326 et seq.: Eyes subjected to saline injection exhibited marginally higher IOPs and lower outflow facilities on average, in comparison to the transgenic animals at baseline. However, due to the lack of statistical significance in these differences and the inherent age difference between the saline-injected animals and the non-injected controls at baseline, no conclusive inference can be drawn regarding the effect of saline injection. To investigate this phenomenon further, we also analyzed IOPs in uninjected contralateral eyes at the midterm time point (Supplementary Tables 1 and 2, Supplementary Figure 4). The uninjected contralateral transgenic eyes (10 months old) showed an IOP of 16.5 [15.9, 17.1] mmHg, which was intermediate between the IOP levels of the 6–7-month-old Tg group (15.4 [14.7, 16.1] mmHg) and the sham-injected group (16.9 [15.5, 18.2] mmHg). However, none of these differences reached statistical significance. Of note, contralateral hypertension has been previously reported after subconjunctival and periocular injection of dexamethasoneloaded nanoparticles (34), and we similarly cannot definitively rule out potential contralateral effects induced by our stem cell injections. Thus, we cannot draw any definite conclusions from these additional IOP comparisons at this time.

**Reviewer #3 (Public review):**
Summary:The purpose of the current manuscript was to investigate a magnetic cell steering technique for efficiency and tissue-specific targeting, using two types of stem cells, in a mouse model of glaucoma. As the authors point out, trabecular meshwork (TM) cell therapy is an active area of research for treating elevated intraocular pressure as observed in glaucoma. Thus, further studies determining the ideal cell choice for TM cell therapy is warranted. The experimental protocol of the manuscript involved the injection of either human adipose derived mesenchymal stem cells (hAMSCs) or induced pluripotent cell derivatives (iPSC-TM cells) into a previously reported mouse glaucoma model, the transgenic MYOCY437H mice and wild-type littermates followed by the magnetic cell steering. Numerous outcome measures were assessed and quantified including IOP, outflow facility, TM cellularity, retention of stem cells, and the inner wall BM of Schlemm's canal.Strengths:All of these analyses were carefully carried out and appropriate statistical methods were employed. The study has clearly shown that the hAMSCs are the cells of choice over the iPSC-TM cells, the latter of which caused tumors in the anterior chamber. The hAMSCs were shown to be retained in the anterior segment over time and this resulted in increased cellular density in the TM region and a reduction in IOP and outflow facility. These are all interesting findings and there is substantial data to support it.Weaknesses:However, where the study falls short is in the MYOCY437H mouse model of glaucoma that was employed. The authors clearly state that a major limitation of the study is that this model, in their hands, did not exhibit glaucomatous features as previously reported, such as a significant increase in IOP, which was part of the overall purpose of the study. The authors state that it is possible that "the transgene was silenced in the original breeders". The authors did not show PCR, western blot, or immuno of angle tissue of the tg to determine transgenic expression (increased expression of MYOC was shown in the angle tissue of the transgenics in the original paper by Zode et al, 2011). This should be investigated given that these mice were rederived. Thus, it is clearly possible that these are not transgenic mice.

All MYOC mice that were used in this study were genotyped and confirmed to carry the transgene as noted in the original version of the paper (see lines 590-2). However, the transgene seems not to have been active, based on the lack of ocular hypertension as well as the lack of differences in supporting endpoints such as outflow facility and TM cellularity. While it would have been possible to carry out their recommended assays to investigate the root cause of this loss of phenotype this was not an objective of our study. Thus we instead here focus simply on communicating the observed loss of phenotype to readers. We also refer the referee to the final paragraph of our response to Referee 1.

If indeed they are transgenics, the authors may want to consider the fact that in the Zode paper, the most significant IOP elevation in the mutant mice was observed at night and thus this could be examined by the authors.

This is a good point. However, while the dark-phase IOP does exhibit a distinctly larger elevation (as previously observed in hypertonic saline sclerosis), Zode et al. also reported a notable 3 mmHg IOP increase during the light phase. The complete absence of such daytime (light phase) IOP elevation in our animals diminished our enthusiasm for pursuing darkphase IOP measurements.

Other glaucomatous features of these mice could also have been investigated such as loss of RGCs, to further determine their transgenic phenotype.

We agree that these other phenotypes could be studied, but in the absence of any detectable IOP elevation (and thus lack of mechanical insult on RGC axons), loss of RGC is extremely unlikely. We also note that the loss of retinal ganglion cells (RGCs) in the Myocilin model remains a subject of controversy. For example, despite a significant increase in IOP (>10 mmHg) in this model across four mouse strains, three, including C57BL6/J, did not exhibit any signs of optic nerve damage (McDowell et al., 2012). In contrast, Zhu et al. observed considerable nerve damage in this model, which was reversed following iPSC-TM cell transplantation (Zhu et al., 2016). Given these conflicting findings, we directed our efforts toward outcome measures directly related to aqueous humor dynamics.

Finally, while increased cellular density in the TM region was observed, proliferative markers could be employed to determine if the transplanted cells are proliferating.

We agree that identifying the source of the increased trabecular meshwork (TM) cellularity we observed is interesting and we plan to pursue that in future studies.

**Recommendations for the authors:**

**Reviewer #1 (Recommendations for the authors):**
The sham-injected transgenic animals showed elevated IOP 3-4 weeks after the baseline measurements in the transgenic mice. The authors justify this may be due to the increase in age in these animals. However, this seems unlikely due to the short duration of time between measurement of the baseline IOP and the Short time point (3-4 weeks). The authors do not provide IOP data for any WT sham injected eyes or naïve Tg eyes at these time points. These data are essential to determine if the elevation is due to the sham injection, age, or the transgene. Could it be that the IOP in this cohort of Tg mice didn't increase until 7-8 months of age instead of 6-7 months of age? The methods state only unilateral injections of the stem cells were done so it is assumed the contralateral eye was uninjected. What was the IOP in these eyes? These data would clarify the confusion in the data from sham-injected animals compared to baseline (naive) measurements.

We agree that the average IOP in saline-injected groups is higher than in WT or non-treated Tg mice, although the difference is inconclusive due to a lack of statistical significance. It is important to note, however, that this difference is subtle and not comparable to the 3 mmHg light-phase IOP elevation previously observed in this model (Zode et al., 2011).

We appreciate the reviewer’s suggestion to include IOP data from the contralateral uninjected eyes, and we have now provided this information along with the comparative statistics in the supplementary materials. Additional details can be found in our response to a similar comment from Reviewer 2’s public review. In summary, the IOP difference in contralateral non-injected ten-month-old transgenic eyes was even smaller than in the original Tg group. IOP elevation following saline injection in mice has been reported previously (Zhu et al., 2016). As a potential confounding factor, we highlight possible contralateral effects of the injection itself (which is why we initially did not analyze IOP in the contralateral eyes).

The hAMSC-treated eyes appear to lower IOP even from baseline (although stats were only provided compared to the sham-injected eyes, which as stated above appear to have increased).However, the iPSC-TM-treated eyes had IOPs equal to that of the baseline measurements taken 3 weeks prior. The significance is coming from the "sham-treated" eyes which had elevated IOPs. The controls listed above should be included to make these conclusions.

The reviewer makes an astute observation. Please refer to our response to a similar observation by Reviewer 2 under public reviews, where we provide and discuss the comparative statistics noted by the reviewer. However, we feel very strongly that the most relevant IOP comparison is between cell-injected eyes and control-injected eyes.

If the transgenic mouse model truly did not have a phenotype, then the authors are testing the ability of the stem cells to lower IOP from baseline normal pressures. Therefore, the authors are not "restoring function of the conventional outflow pathway" as there is no damage to begin with. The language in the manuscript should be corrected to reflect this if the transgenics have no phenotype.

We agree and have adjusted the language accordingly. For further details, please refer to our response to your public review.

The authors noted in the iPSC-TM-treated eyes there was a high rate of tumorigenicity. If the magnetic steering of these cells is specific and targeted to the TM, why do the tumors form near the central iris?

While magnetic steering is more specific to the trabecular meshwork (TM) than previously used approaches (Bahrani Fard et al., 2023), it is not perfect, and a modest amount of offtarget delivery to the iris, including its central portion, still occurs. Apparently, it took only a few mis-directed iPSC-TM cells to lead to tumors in this work, which is a serious concern for future translational approaches.

**Reviewer #2 (Recommendations for the authors):**
(1) It appears that mice were injected unilaterally (Line 590). I may have missed this, but was the companion un-injected eye analyzed in this study? If not analyzed, was there a confounding concern or limitation that necessitated omitting this possible control option?

Contralateral effects, such as hypertension in the untreated eye after subconjunctival and periocular injection of dexamethasone-loaded nanoparticles, have previously been reported in the literature (Li et al., 2019) and also reported anecdotally by other leaders in the field to the senior authors, which is why we did not initially analyze contralateral eyes in this study. However, prompted by this comment and others, we have now included the IOP measurements for contralateral uninjected ten-month-old transgenic eyes in the supplementary materials. For further details, please refer to our response to your public review.

(2) Were all these mice the same gender? Would gender be expected to alter the findings of this study?

Animals of both sexes were randomly chosen and included in the study. We added the following statement to the Materials and Methods section (line 530): After breeding and genotyping, mice, regardless of sex, were maintained to age 6-7 months, when transgenic animals were expected to have developed a POAG phenotype.

(3) As noted in the public review, the use of PBS for a control seems to have resulted in a slight elevation in IOP (Figure 2) as well as a reduction in outflow facility (Figure 3B) when compared to WT or Tg mice. Was this difference statistically significant?

The differences between the sham (saline)-injected groups at any time point and untreated Tg mice did not reach statistical significance for IOP, facility, or TM cellularity and for facility, did not even show clear trends. For example, WT mice had, on average, 0.2 mmHg higher IOP and 0.6 nl/min/mmHg greater facility than the Tg group. Meanwhile on a similar scale, the long-term sham group exhibited 0.4 nl/min/mmHg higher facility compared to the Tg group. As the statistical tests indicate, these differences should be interpreted more as noise than meaningful signal.

If so, then it should be noted as to whether the observed decrease in IOP following stem cell injection remained statistically significant when compared to these un-injected control animals. If significance was lost, then this should be appropriately noted and discussed. It is not apparently obvious why sham controls should have elevated IOP. This is a design and statistical concern.

Please refer to our response to a similar observation by Reviewer 1. We believe that comparing the treatment (cell suspension in saline) with its age-matched vehicle (saline) is the appropriate approach which maintains rigor by most directly accounting for the effects of injection.

(4) The tonicity of the PBS used as a vehicle control was not stated and I did not see within the methods whether the stem cells were suspended using this same PBS vehicle. I assume isotonic phosphate buffered saline was used and that the stem cells were resuspended using the same sterile PBS.

Thanks for catching this. We added “sterile PBS (1X, Thermo Fisher Scientific, Waltham, MA)” to the Methods section of the manuscript (line 567).

With regards to using PBS as an injection control, I wonder if a better comparable control might have been to use mesenchymal stem cells that were rendered incapable of proliferating prior to intracameral injection. This, of course, addresses the unexplained mechanism(s) by which mesenchymal stem cells elicit a decrease in IOP.

This is an interesting idea, and represents another level of control. However, we explicitly chose not to use non-proliferating hAMSCs as a control, for several reasons. Firstly, a saline injection is the simplest control and in this initial study with multiple groups, we did not feel another experimental group should be added. Second, this control would not rule out paracrine effects from injected cells, which our data suggested are an important effect. Third, rendering injected cells truly non-proliferative could introduce unwanted/unknown phenotypes in these cells that would need to be carefully characterized. That being said, if an efficient method could be developed to render an entire population of these cells irreversibly non-proliferating, the reviewer’s suggestion would be worth pursuing to better understand the mechanism of TM cell therapies.

(5) As noted in Figure 4C, TM cellular density as quantified was not altered in the sham control, so a loss of cellular density can not explain the elevated IOP with this group. Injecting viable (not determined?) mesenchymal stem cells did show, over the short term, a noted increase in TM cellular density.

Thank you for noting this. We agree that changes in cell density do not explain the mild IOP elevation in the sham group. As the referee certainly is aware, there are multiple reasons that IOP can be elevated (changes in trabecular meshwork extracellular matrix, changes in trabecular meshwork stiffness) that are not necessarily related to cell density. Since we do not know definitively the cause of this mild elevation, we would prefer to not speculate about it in the manuscript.

Thanks for pointing out our omission of a statement about injected cell viability. We have now included the following statement in the Materials and Methods section (564-566): “For all the experiments where animals received hAMSC, cell count and >90% viability was verified using a Countess II Automated Cell Counter (Thermo Fisher Scientific, Waltham, MA).”

I'm confused, as clearly stated (Lines 431-432), mesenchymal stem cells accumulated close to, but not within, the TM. How is it that TM cellular density increased if these stem cells did not enter the TM? The authors may wish to clarify this distinction. Given that mesenchymal stem cells did not increase the risk of tumorigenicity, do the authors have any evidence that these cells actually proliferated post-injection or did they undergo senesce thereby displaying senescence-associated secretory phenotype as a source of paracrine support?

As the reviewer correctly noted, our observations show that hAMSCs primarily accumulated close to, but outside, the TM (likely caught up in the pectinate ligaments). Based on observations of increased TM cellularity, we think that the most likely explanation of these findings is paracrine signaling, as the reviewer suggests and which was discussed at length in the original version of the manuscript (lines 453-477).

We agree that, despite observing little signal from hAMSCs within the TM, labeling with proliferation markers (e.g., Ki-67) and searching for co-localization with exogenous cells, and/or labeling for senescence markers would have provided more mechanistic information. This is an excellent topic for future study, which we plan to pursue, but was outside the scope of this study.

(6) As noted in the public review, I think it is a bit of a stretch to even suggest that the findings of this study support stem cell restoration of TM function given that the model apparently did not produce TM cell dysfunction as anticipated. A restoration effect remains to be seen.

We agree and have adjusted the language accordingly. For further details, please refer to our response to Reviewer 1’s public comment.

Reviewer #3 (Recommendations for the authors):(1) Show PCR, western blot, or immuno of angle tissue of the MYOC tg to confirm transgenic expression.(2) Examine the IOP of mice at night.(3) Investigate other glaucomatous features in the mice to determine if they have any of the transgenic phenotypes previously reported.(4) Examine proliferative markers in the TM region of angles injected with stem cells.

Please see our responses to all four of these comments in the public section.

Bibliography (for this response letter only)

Bahrani Fard, M.R., Chan, J., Sanchez Rodriguez, G., Yonk, M., Kuturu, S.R., Read, A.T., Emelianov, S.Y., Kuehn, M.H., Ethier, C.R., 2023. Improved magnetic delivery of cells to the trabecular meshwork in mice. Exp. Eye Res. 234, 109602. https://doi.org/10.1016/j.exer.2023.109602

Li, G., Lee, C., Agrahari, V., Wang, K., Navarro, I., Sherwood, J.M., Crews, K., Farsiu, S., Gonzalez, P., Lin, C.-W., Mitra, A.K., Ethier, C.R., Stamer, W.D., 2019. In vivo measurement of trabecular meshwork stiffness in a corticosteroid-induced ocular hypertensive mouse model. Proc. Natl. Acad. Sci. U. S. A. 116, 1714–1722.

https://doi.org/10.1073/pnas.1814889116

Zhu, W., Gramlich, O.W., Laboissonniere, L., Jain, A., Sheffield, V.C., Trimarchi, J.M., Tucker, B.A., Kuehn, M.H., 2016. Transplantation of iPSC-derived TM cells rescues glaucoma phenotypes in vivo. Proc. Natl. Acad. Sci. 113, E3492–E3500.

Zode, G.S., Kuehn, M.H., Nishimura, D.Y., Searby, C.C., Mohan, K., Grozdanic, S.D., Bugge, K., Anderson, M.G., Clark, A.F., Stone, E.M., Sheffield, V.C., 2011. Reduction of ER stress via a chemical chaperone prevents disease phenotypes in a mouse model of primary open angle glaucoma. J. Clin. Invest. 121, 3542–3553. https://doi.org/10.1172/JCI58183